# Curvature-Aware Safety Restoration In LLMs Fine-Tuning

**Thong Bach**                                                       *t.bach@deakin.edu.au*
*Applied Artificial Intelligence Initiative (A2I2)*
*Deakin University*

**Thanh Nguyen-Tang**
*Department of Data Science*
*New Jersey Institute of Technology*

**Dung Nguyen**
*Applied Artificial Intelligence Initiative (A2I2)*
*Deakin University*

**Thao Minh Le**
*Pennsylvania State University*

**Truyen Tran**
*Applied Artificial Intelligence Initiative (A2I2)*
*Deakin University*

**Reviewed on OpenReview:** *https://openreview.net/forum?id=FSUehLhGyl*

## Abstract

Fine-tuning Large Language Models (LLMs) for downstream tasks often compromises safety alignment, even when using parameter-efficient methods like LoRA. In this work, we uncover a notable property: fine-tuned models preserve the geometric structure of their loss landscapes concerning harmful content, regardless of the fine-tuning method employed. This suggests that safety behaviors are not erased but shifted to less influential regions of the parameter space. Building on this insight, we propose a curvature-aware alignment restoration method that leverages influence functions and second-order optimization to selectively increase loss on harmful inputs while preserving task performance. By navigating the shared geometry between base and fine-tuned models, our method discourages unsafe outputs while preserving task-relevant performance, avoiding full reversion and enabling precise, low-impact updates. Extensive evaluations across multiple model families and adversarial settings show that our approach efficiently reduces harmful responses while maintaining or even improving utility and few-shot learning performance.

# 1 Introduction

Large Language Models (LLMs) encode safety-aligned behaviors during pretraining, but these safeguards deteriorate during task-specific fine-tuning, a phenomenon we identify as *safety alignment drift*. Studies demonstrate that even minimal fine-tuning can compromise safety mechanisms, with models like GPT-3.5 Turbo becoming consistently unsafe after adaptation on just 10 adversarial examples Qi et al. (2023b); Bach et al. (2026). Attempts to address this issue by modifying model behavior generally fall into two main categories, both of which suffer from inherent limitations. **Behavioral unlearning** methods attempt to remove undesirable knowledge or responses (Cao & Yang, 2015; Bourtoule et al., 2021a), but often require costly retraining or risk catastrophic forgetting. **Model editing** approaches aim to update factual associations or local behaviors through direct parameter intervention (Meng et al., 2022; Mitchell et al., 2022), yet struggle to generalize beyond narrow scopes or isolated prompts. To solve these issues, we propose a new direction that treats safety behavior as an intrinsic property of the model's geometry and seeks to restore alignment through curvature-aware navigation of the loss landscape.

Our key insight, supported by extensive empirical analysis (Section 2), is that models preserve notable structural properties in their loss landscapes with respect to harmful content after finetuning. Specifically, we observe high correlations in models' responses to harmful inputs before and after fine-tuning, despite substantial divergence in other functional behaviors. This suggests that safety mechanisms remain largely preserved in the parameter space, merely shifted to less dominant regions during task-specific optimization.

This observation motivates our novel approach: *curvature-aware alignment restoration.* We leverage the preserved geometry of the loss landscape to restore safety boundaries. By employing influence functions and second-order optimization techniques, our method navigates the parameter space to increase loss on harmful inputs while minimizing impact on task performance. Our contributions include:

- We identify and empirically validate a key insight: Fine-tuning preserves the geometric structure of the loss landscape for harmful content across diverse model families.

- We propose a curvature-aware alignment restoration method that leverages influence functions and second-order optimization to suppress harmful behaviors.

- We demonstrate that our approach significantly reduces harmful responses while preserving task performance, enhancing few-shot generalization, and improving robustness to adversarial attacks and parameter perturbations.

# 2 Empirical Evidence and Loss Landscape Analysis

In this section, we first present empirical evidence demonstrating high correlations between base and fine-tuned models' responses to harmful content across both parameter-efficient and full finetuning methods, despite divergence in task performance. We then visualize and quantify this preserved geometry through loss landscape analysis, providing the foundation for our curvature-aware restoration approach.

Table 1: Pearson correlation coefficients between base and fine-tuned models' responses across harmful content (HEx-PHI), task-specific data (Dolly), general data (Alpaca), and domain-specific datasets (CodeAlpaca, MedMCQA, SquAD v2 ). Harmful content exhibits consistently high correlations (>0.85) across both LoRA and full fine-tuning, while domain-specific data shows highly variable correlations (−0.47 to 0.86). This asymmetric preservation, consistent structure for harmful content but variable structure for other domains, rules out simple out-of-distribution effects and validates our hypothesis that safety mechanisms occupy a functionally distinct region in the loss landscape.

| Fine-tuning | Models | Harmful | Dolly | Alpaca | CodeAlpaca | MedMCQA | SquAD v2 |
|---|---|---|---|---|---|---|---|
| | LLaMA-2 7B | **0.992** | 0.056 | −0.055 | 0.551 | −0.465 | 0.596 |
| **LoRA** | LLaMA-3.1 8B | **0.995** | 0.550 | 0.510 | 0.780 | 0.554 | 0.516 |
| | Qwen 2.5 7B | **0.994** | 0.014 | 0.067 | 0.696 | 0.862 | 0.515 |
| | LLaMA-2 7B | **0.852** | −0.004 | 0.185 | 0.396 | 0.597 | 0.649 |
| **Full FT** | LLaMA-3.1 8B | **0.990** | 0.535 | 0.508 | 0.746 | 0.499 | 0.366 |
| | Qwen 2.5 7B | **0.941** | 0.526 | 0.129 | 0.363 | 0.012 | 0.312 |

## 2.1 Empirical Validation

We analyze multiple model families, measuring Pearson correlation coefficients between base and tuned models' responses across diverse data categories: harmful content (HEx-PHI Qi et al. (2023b): a benchmark dataset of 330 harmful instructions across 11 policy-based categories), task-specific data (Dolly testset Databricks (2023), 200 examples), general data (Alpaca testset Taori et al. (2023), 200 examples), and domain-specific datasets including code generation (CodeAlpaca), medical reasoning (MedMCQA (Pal et al., 2022)), and question answering (SquAD v2 (Rajpurkar et al., 2018)). We evaluate both LoRA and full fine-tuning to verify that our findings generalize across fine-tuning methodologies.

These correlations quantify how consistently models respond to the same inputs before and after fine-tuning. For each dataset $\mathcal{D}$, we compute the Pearson correlation coefficient:

$$r = \frac{\sum_{x \in \mathcal{D}}(L_{\text{base}}(x) - \overline{L}_{\text{base}})(L_{\text{tuned}}(x) - \overline{L}_{\text{tuned}})}{\sqrt{\sum_{x \in \mathcal{D}}(L_{\text{base}}(x) - \overline{L}_{\text{base}})^2}\sqrt{\sum_{x \in \mathcal{D}}(L_{\text{tuned}}(x) - \overline{L}_{\text{tuned}})^2}}$$

where $L_{\text{base}}(x)$ and $L_{\text{tuned}}(x)$ are the cross-entropy losses of the base and fine-tuned models on example $x$, and $\overline{L}_{\text{base}}$ and $\overline{L}_{\text{tuned}}$ are their respective mean values across dataset $\mathcal{D}$. Higher correlation indicates the fine-tuned model maintains similar response behavior to the base model, despite parameter changes. By comparing correlations across different input categories, we can detect whether safety-relevant behaviors remain intact despite changes to task-specific capabilities.

Our analysis reveals three key insights:

1. **Consistent safety structure preservation across fine-tuning methods:** In Table 1, harmful content shows consistently high correlation (0.85–0.99) across both LoRA and full fine-tuning methods and all three model families. Full fine-tuning preserves loss structure slightly less than LoRA (e.g., 0.852 vs 0.992 for LLaMA-2 7B), which aligns with intu-

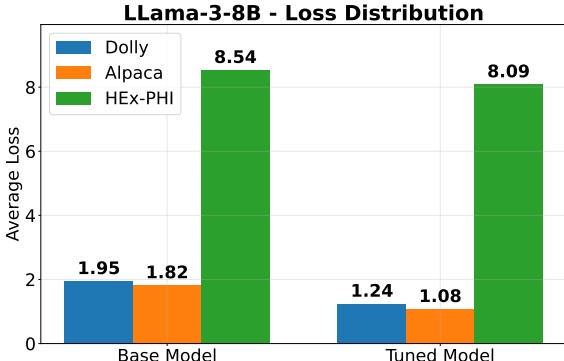

Figure 1: Average loss comparison across base and fine-tuned LLama-3 8B Instruct models for three datasets: Dolly (task-specific), Alpaca (general), and HEx-PHI (harmful). Harmful content consistently exhibits higher loss compared to benign content in both model states, showing that harmful content consistently lies in a distinct and preserved region of the loss landscape.

   ition since full fine-tuning modifies all parameters rather than a low-rank subspace, yet preservation remains strong ($\geq 0.85$).

2. **Variable structure for domain-specific data:** In contrast, domain-specific datasets exhibit highly variable correlations ranging from $-0.47$ to $0.86$ across models and fine-tuning methods. Although some datasets show moderate correlations (e.g., CodeAlpaca 0.78 for LLaMA-3.1, MedMCQA 0.86 for Qwen), this variability contrasts sharply with the consistently high correlations for harmful content (0.85 to 0.99). If preservation were merely an out-of-distribution effect, all dissimilar data would show similarly high correlations. The observed variability confirms that safety behaviors occupy a functionally distinct region rather than reflecting distributional distance alone.

3. **Distinct safety regions in loss landscape:** In Figure 1 we measure the loss of LLaMA-3.1-8B-Instruct on these data. Generally, harmful content consistently generates higher loss values (8.54 and 8.09) compared to benign content (1.82–1.95 and 1.08–1.24) in both model states, suggesting potential separation between harmful and task-relevant regions in the loss landscape. More detailed loss analysis will be presented in Appendix C.5.

Based on these findings, we state our hypothesis: **safety behaviors exist in a functionally distinct region of the loss landscape that remains largely undisturbed by task-specific fine-tuning, regardless of whether parameter-efficient or full fine-tuning is employed**. Therefore, developing a targeted restoration method to recover safety behaviors without compromising useful task capabilities is feasible.

## 2.2 Loss Landscape visualization

To further support our hypothesis, we visualize the loss landscapes of both the base and fine-tuned models using a 3D projection technique. Rather than sampling arbitrary directions in parameter space, we construct perturbation directions informed by gradients computed on harmful and benign

inputs. Specifically, we focus on attention and MLP layers, which most strongly influence model behavior. For each model, we generate two approximately orthogonal perturbation vectors ($\mathbf{d}_1$ and $\mathbf{d}_2$) and evaluate the model's loss across a grid ($20 \times 20$) of perturbation magnitudes. We create this grid by varying coefficients $\lambda_1$ and $\lambda_2$ within the range $[-0.01, 0.01]$ and applying the perturbation $\theta_{perturbed} = \theta_{original} + \lambda_1 \mathbf{d}_1 + \lambda_2 \mathbf{d}_2$ to the model parameters. At each grid point, we compute the loss using a consistent set of 32 validated samples, resulting in a 3D surface where the $x$-axis and $y$-axis represent perturbation magnitudes along each direction, and the $z$-axis shows the corresponding loss value. Full implementation details are provided in Appendix B.3.

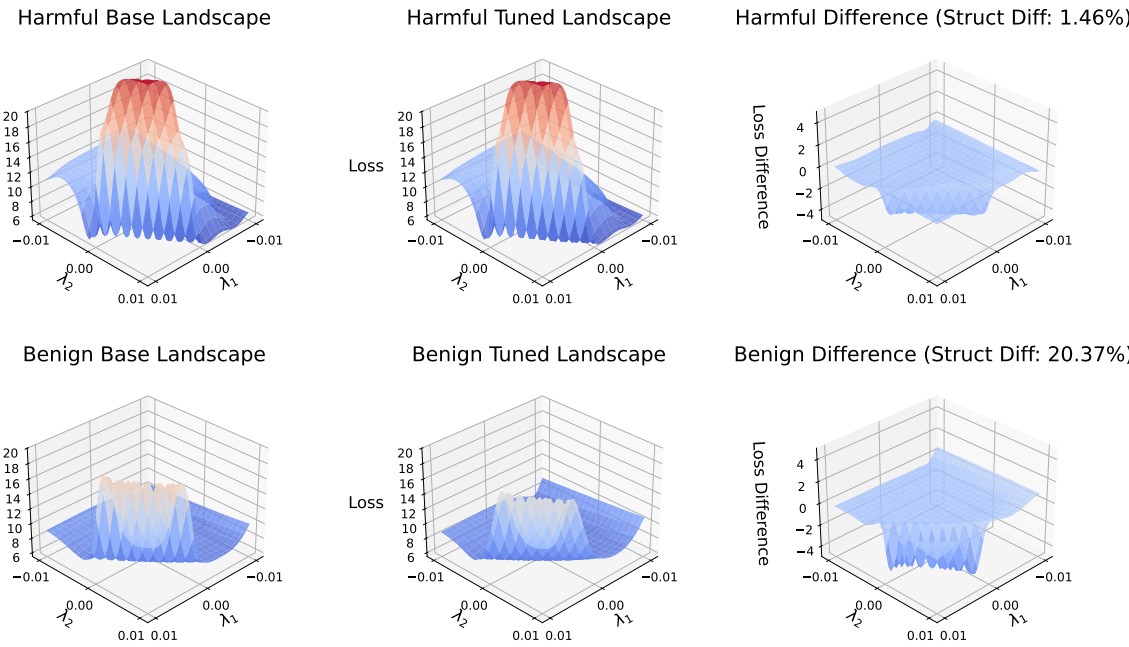

Figure 2: 3D loss landscape visualization for LLama-3 8B Instruct using gradient-informed direction projection (Section 2.2). The top row shows the loss landscape of harmful content (HEx-PHI), while the bottom row shows for general data (Alpaca). Comparison between base (left) and fine-tuned (middle) models reveals preserved topological features for harmful content (structural difference: 1.46%), while general data landscapes undergo substantial transformation (structural difference: 20.37%) . These quantitative measures of landscape change confirm that safety-relevant regions remain largely undisturbed during task-specific fine-tuning. Detailed percentile statistics are provided in Appendix C.7.

The 3D plot in Figure 2 reveals clear evidence for our hypothesis. The loss landscape for harmful content maintains remarkably similar topological features between base and fine-tuned models, with consistent valleys, peaks, and curvature characteristics. We quantify this structural preservation using a correlation-based metric: StructDiff $= (1 - |\text{corr}(\nabla^2 L_{\text{base}}, \nabla^2 L_{\text{tuned}})|) \times 100\%$, where $\nabla^2 L$ is the Laplacian of the loss landscape. This metric captures differences in curvature patterns rather than absolute loss values, revealing only 1.46% structural difference for harmful content despite fine-tuning.

In contrast, the loss landscape for general-purpose data changes significantly, exhibiting 20.37% structural difference in both global geometry and local minima positions. This visualization provides direct confirmation that fine-tuning primarily affects task-specific regions of the parameter space while leaving safety-relevant regions structurally maintained, creating a natural opportunity for targeted safety restoration. Additional results on loss landscapes for LoRA fine-tuning are available in the Appendix B.3.

These visualization results explain the high correlation coefficients documented in Section 2.1 and establish a foundation for our alignment restoration approach. The idea is to identify and leverage preserved landscape features and use these to navigate toward parameter configurations that maintain task performance while reinforcing safety boundaries.

## 3 Curvature-aware alignment restoration

The preserved loss landscape geometry identified in Section 2.1 has a direct implication: safety restoration can be achieved through small, precise parameter updates rather than large-scale re-training. If fine-tuning had destroyed safety-relevant structure, restoring appropriate refusal behavior would require either reconstructing these mechanisms from scratch or making large parameter changes that risk catastrophic forgetting on the task. The high correlation we observe indicates instead that safety behaviors occupy a functionally distinct region that remains accessible in parameter space. This enables a constrained optimization approach where we can restore safety within tight utility bounds.

We now introduce our curvature-aware alignment restoration approach, which provides a principled way to steer a fine-tuned LLM back toward the safety behavior encoded in its base model while preserving desirable task-specific knowledge.

### 3.1 Problem Formulation and Optimization Approach

Let us define $\theta_{\text{base}}$ as the parameters of the pretrained, safe base model, and $\theta_{\text{tuned}}$ as the parameters of the fine-tuned model. We use two distinct datasets:

a *retain set* containing benign, task-relevant examples where performance should be preserved, and a *forget set* containing potentially harmful examples where safety alignment should be restored.

For both datasets, we employ the standard autoregressive language modeling loss:

$$L(x; \theta) = -\sum_{i=1}^{|x|} \log p_\theta(x_i | x_{<i}) \tag{1}$$

where $x$ represents an input sequence and $p_\theta(x_i|x_{<i})$ is the model's predicted probability for token $x_i$ given preceding tokens.

Our goal is to update $\theta_{\text{tuned}}$ toward a point $\theta_{\text{updated}}$ that preserves $L_{\text{retain}}$ (the loss on retain set) while increasing $L_{\text{forget}}$ (the loss on forget set). We formulate this as a constrained optimization problem:

$$\max_\theta L_{\text{forget}}(\theta) \quad \text{s.t.} \quad L_{\text{retain}}(\theta) \leq L_{\text{retain}}(\theta_{\text{tuned}}) + \epsilon \tag{2}$$

where $\epsilon$ is a small positive scalar allowing limited degradation in retain set performance. Based on extensive empirical validation, we established $\epsilon = 0.1$ as a default constraint threshold, ensuring the recovered model maintains task performance within an acceptable margin of the fine-tuned baseline.

This formulation can be theoretically justified through a second-order Taylor approximation of the retain loss around $\theta_{\text{tuned}}$:

$$L_{\text{retain}}(\theta_{\text{tuned}} + \Delta\theta) \approx L_{\text{retain}}(\theta_{\text{tuned}}) + \nabla L_{\text{retain}}^{\top}\Delta\theta + \frac{1}{2}\Delta\theta^{\top}H_{\text{retain}}\Delta\theta \tag{3}$$

Under this approximation, the influence function update provides the steepest descent direction for $L_{\text{forget}}$ in the Riemannian geometry defined by $H_{\text{retain}}$ Amari (1998):

$$\Delta\theta_{\text{influence}} = \arg\max_{\Delta\theta} \nabla L_{\text{forget}}^{\top}\Delta\theta \quad \text{s.t.} \quad \|\Delta\theta\|_{H_{\text{retain}}} \leq \delta \tag{4}$$

Here, $\delta > 0$ defines the allowable trust region radius with respect to the local geometry of the retain loss, measured via the Mahalanobis norm $\|\Delta\theta\|_{H_{\text{retain}}} = \sqrt{\Delta\theta^{\top}H_{\text{retain}}\Delta\theta}$. This parameter is directly related to the constraint threshold $\epsilon$ in Equation 2: smaller values of $\delta$ ensure updates remain in regions where the quadratic approximation is valid, thereby helping satisfy the $\epsilon$-bounded retain loss constraint. Intuitively, this constraint ensures that the update direction increases the forget loss without significantly increasing the retain loss, as measured by its local curvature. Solving this constrained optimization yields the steepest ascent direction for $L_{\text{forget}}$ under a Riemannian metric induced by $H_{\text{retain}}$.

Directly solving Equation 4 may be computationally expensive. Therefore, we adopt a tractable approximation based on influence functions, as shown below:

$$\Delta\theta_{\text{influence}} = H_{\text{retain}}^{-1}\nabla L_{\text{forget}}(\theta_{\text{tuned}}) \tag{5}$$

This approximation can be interpreted as the unconstrained solution to Equation 4, where the trust region constraint is relaxed. Specifically, Equation 5 represents the steepest ascent direction for $L_{\text{forget}}$ under the curvature geometry of the retain set, without explicitly enforcing a norm constraint. However, since we have removed the explicit trust region constraint, we need to compensate by adding practical safeguards. We achieve this through step scaling (controlling update magnitudes) and L-BFGS curvature filtering (ensuring numerical stability), as detailed in Appendix B.1.

In practice, we construct $H_{\text{retain}}^{-1}$ using a low-rank L-BFGS Liu & Nocedal (1989) approximation that incorporates curvature information from both the retain set and a subset of the forget set. This hybrid construction enables the trust region to balance retention of task-specific knowledge with awareness of harmful content boundaries, resulting in more effective influence updates. We discuss implementation details and ablation results in the Appendix B.1.

### 3.2 Practical Implementation

Directly computing and inverting the Hessian matrix $H_{\text{retain}}$ for modern LLMs is computationally intractable due to the enormous parameter space. To address this challenge, we implement two key techniques:

**(1) Parameter-Efficient Fine-Tuning.** We apply our method within the Low-Rank Adaptation (LoRA) framework. This reduces the dimensionality of the Hessian matrix to just the trainable parameters, making curvature estimation feasible.

**(2) Approximate Hessian Inversion.** We employ L-BFGS (Limited-memory Broyden–Fletcher–Goldfarb–Shanno) to efficiently approximate $H_{\text{retain}}^{-1}$, reducing computation to $\mathcal{O}(mp)$ where $m$ is the memory size and $p$ is the parameter dimensionality.

This quasi-Newton method builds an approximation of the inverse Hessian through successive low-rank updates, avoiding explicit matrix inversion.

## 4 Experimental Results

In this section, we empirically evaluate our curvature-aware safety restoration method across diverse architectures and benchmarks. Our experiments investigate three core questions: **(1)** How well does our method restore safety compared to state-of-the-art approaches? **(2)** Does it preserve model utility and adaptability? **(3)** How robust is the restored alignment to adversarial attacks and parameter perturbations? We outline our experimental setup—architectures, fine-tuning protocol, and baselines—before presenting results on safety performance, utility preservation, in-context learning, and robustness to both prefilling attacks and weight-space perturbations. Overall, our method consistently restores safety without degrading task performance, addressing a central challenge in fine-tuning LLMs.

### 4.1 Experimental Setup

**Base LLMs** We evaluate our curvature-aware alignment restoration method on three representative large language models that span different architectures and training paradigms: LLama-2 7B Chat, LLama 3.1 8B Instruct, and Qwen 2.5 7B Instruct. These models were selected for their widespread adoption in the research community, comparable parameter scales (7-8B parameters), which allow us to assess how our method generalizes across model families with different inherent safety characteristics.

**Fine-tuning Protocol** To maintain computational efficiency while preserving model quality, we implement Parameter-Efficient Fine-Tuning (PEFT) via Low-Rank Adaptation (LoRA). Across all experiments, we utilize a consistent configuration with rank $r = 32$ and learning rate $\alpha = 2 \times 10^{-4}$. We apply LoRA adapters to the query and value projections in attention layers, following the default configuration used in the PEFT library Mangrulkar et al. (2022).

For our primary instruction-tuning dataset, we employ Dolly, a diverse collection of 15,000 human-generated instruction-response pairs spanning multiple domains. We fine-tune each model for 1

epoch with a batch size of 128 examples, using the AdamW optimizer. All experiments were conducted on 1 NVIDIA H100 GPUs with 80 GB memory.

**Baseline Methods** We compare our curvature-aware alignment restoration approach against several state-of-the-art methods for safety-preserving fine-tuning:**(1) Vanilla Fine-tuning** Hu et al. (2022): Standard LoRA fine-tuning without any safety preservation mechanisms, serving as our primary control. **(2) Vaccine** Huang et al. (2024): A preventative approach that operates during the initial alignment phase by adding crafted perturbations to hidden embeddings, making the model robust against harmful perturbations that may be introduced during subsequent fine-tuning. **(3) Safe LoRA** Hsu et al. (2024): A data-free, training-free approach that preserves safety alignment during fine-tuning by projecting LoRA weight updates onto an alignment subspace defined by the difference between aligned and unaligned model weights, applying this projection only when updates deviate significantly from the alignment direction. **(4) SaLoRA** Li et al. (2025a): A technique that preserves safety alignment during LoRA fine-tuning by introducing a fixed safety module that projects new features to a subspace orthogonal to original safety features, along with task-specific initialization for trainable parameters.

For all baseline methods, we follow the hyperparameter settings recommended in their respective papers, adapting only when necessary to maintain fairness in the comparison.

## 4.2 Safety Evaluation

We evaluate model safety on AdvBench, containing 520 adversarial prompts designed to elicit unsafe responses. We allocate 138 samples for constructing the safety matrix required by SaLoRA and reserve the remaining 382 samples for evaluation. Our primary safety metric is the *harmful response rate* (HRR), calculated as the percentage of evaluation samples eliciting unsafe responses. For a comprehensive assessment, we employ both LLama-3 Guard as an automated safety evaluator and human review to validate the quality and accuracy of safety judgments, ensuring a more reliable evaluation of model safety across different methods.

**Safety performance** Table 2 demonstrates our curvature-aware alignment restoration method achieves superior safety results across model families. For Llama-3.1 8B, our approach reduces HRR to just 3.0%, significantly outperforming both SaLoRA (8.1%), Vaccine (21.3%), and Safe LoRA (11.0%). For Qwen 2.5 7B, we achieve a remarkable 1.5% HRR, substantially lower than all fine-tuning methods including SaLoRA (3.4%). With Llama-2 7B, our method successfully restores complete safety alignment (0% HRR), matching the excellent performance of SaLoRA and Safe LoRA on this model.

**Task Performance and Utility Evaluation.** To show that safety improvements do not compromise task performance, we evaluate models on both the original fine-tuning task (Dolly) and four diverse zero-shot tasks: ARC-Challenge (commonsense reasoning), GSM8K (mathematical reasoning), ToxiGen (toxicity detection), and TruthfulQA (factual consistency). The column 'Eval' in Table 2 shows that our method maintains a comparable performance to other safety techniques in the original fine-tuning task, with scores of 1.3, 1.4, and 1.2 in the three model families.

For broader utility, our approach maintains strong performance across tasks. On Llama-3.1 8B, our method achieves the highest scores on GSM8K (76.5) and TruthfulQA (43.6) while maintain-

Table 2: Comparison of safety restoration methods across three model families. HRR (Harmful Response Rate, lower is better) measures safety on AdvBench, while Eval shows performance on fine-tuning dataset (average cross-entropy loss across all examples in the Dolly test set). Utility metrics include four zero-shot tasks: ARC-Challenge (ARC-C), GSM8K, ToxiGen, and TruthfulQA. Our curvature-aware approach achieves best safety across all models while maintaining competitive task performance. Bold indicates best method, underline indicates second-best for each metric within model family.

| Models | Methods | Eval ↓ | HRR ↓ | Utility ↑ | | | |
|---|---|---|---|---|---|---|---|
| | | | | ARC-C | GSM8K | ToxiGen | TruthfulQA |
| Llama-3.1 8B | Base | 1.9 | 1.4 | 52.0 | 75.2 | 53.3 | 45.5 |
| | LoRA | **1.2** | 25.5 | 51.2 | 72.4 | 44.9 | 39.0 |
| | Vaccine | 1.3 | 21.3 | 44.3 | 39.5 | 43.4 | 34.1 |
| | SaLoRA | **1.2** | 8.1 | **52.3** | 75.7 | **49.3** | 41.8 |
| | Safe LoRA | 1.3 | 11.0 | 51.1 | 75.6 | 48.7 | 42.0 |
| | **Ours** | 1.3 | **3.0** | 51.8 | **76.5** | 46.0 | **43.6** |
| Qwen 2.5 7B | Base | 3.6 | 0.0 | 53.0 | 76.4 | 57.2 | 56.3 |
| | LoRA | **1.2** | 24.7 | **55.0** | 60.2 | 57.2 | 44.5 |
| | Vaccine | **1.2** | 19.3 | 54.6 | 74.3 | **57.9** | 44.5 |
| | SaLoRA | **1.2** | 3.4 | **55.0** | 69.5 | 57.2 | 49.2 |
| | **Ours** | 1.4 | **1.5** | 54.2 | **75.1** | 57.1 | **53.3** |
| Llama-2 7B | Base | 2.5 | 0.0 | 43.3 | 20.1 | 52.9 | 37.2 |
| | LoRA | **1.1** | 21.4 | 44.4 | 19.6 | 44.7 | 32.3 |
| | Vaccine | **1.1** | 16.7 | 42.6 | 11.6 | 41.1 | 31.7 |
| | SaLoRA | **1.1** | **0.0** | **45.9** | **23.6** | 49.5 | 34.7 |
| | Safe LoRA | 1.2 | **0.0** | 45.6 | 21.5 | 43.8 | 33.1 |
| | **Ours** | 1.3 | **0.0** | 44.7 | 22.1 | **51.7** | **36.8** |

ing competitive ARC-C performance (51.8). For Qwen 2.5 7B, we obtain the best performance on TruthfulQA (53.3) and GSM8K (75.1). With Llama-2 7B, our approach achieves the highest TruthfulQA (36.8) and ToxiGen (51.7) scores. This demonstrates our curvature-aware method effectively balances safety restoration with preservation of diverse reasoning capabilities. We further validate in Appendix C.2 that our method generalizes to full fine-tuning, achieving comparable safety restoration (HRR reductions of 57–75%) while maintaining task performance, confirming that the preserved loss landscape geometry enables effective restoration regardless of fine-tuning methodology.

## 4.3 Robustness Evaluation

We evaluate the robustness of our safety alignment restoration through two distinct experiments: resistance to adversarial prefilling attacks and stability under parameter perturbations.

### 4.3.1 Prefilling Attack Resistance

This experiment assesses the robustness of our method against inference-time attacks that exploit shallow safety alignment vulnerabilities in LLMs Qi et al. (2024).

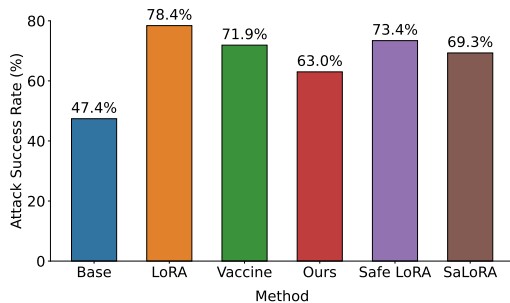

Figure 3: Attack success rates (the lower the better) for prefilling attacks across different alignment restoration methods on Llama-3.1 8B evaluated on AdvBench. Our curvature-aware approach achieves 63.0% ASR, significantly outperforming baseline LoRA (78.4%) and other safety methods, while approaching the robustness of the base model (47.4%).

Table 3: VISAGE scores measuring safety basin robustness. Higher scores indicate more robust safety basins resistant to parameter perturbations. Our approach achieves 56.1, substantially outperforming all baselines.

| Method | VISAGE Score |
|--------|--------------|
| LoRA | 21.1 |
| Vaccine | 28.8 |
| SaLoRA | 32.1 |
| **Ours** | **56.1** |

**Experimental Setup** We use 382 adversarial prompts from AdvBench (used in Section 4.2) to simulate a prefilling attack. Following prior work Qi et al. (2024); Andriushchenko et al. (2024), each input is prepended with four non-refusal tokens, which are designed to bypass the model's standard safety refusal mechanisms.[1]

We evaluate models fine-tuned with five different methods: vanilla LoRA, Vaccine Huang et al. (2024), Safe LoRA Hsu et al. (2024), SaLoRA Li et al. (2025a), and our curvature-aware approach. We report **attack success rate (ASR)** as the percentage of inputs that lead to harmful completions (lower is better).

**Results Analysis** As shown in Figure 3, our method achieves a lower ASR (63.0%) than all other alignment restoration baselines. Compared to standard LoRA fine-tuning (78.4%), our approach yields a 19.6% relative reduction in attack success, and also demonstrates improved robustness over Vaccine, Safe LoRA, and SaLoRA. These findings highlight the effectiveness of our curvature-aware approach in mitigating shallow alignment vulnerabilities and preserving safety under adversarial prompting.

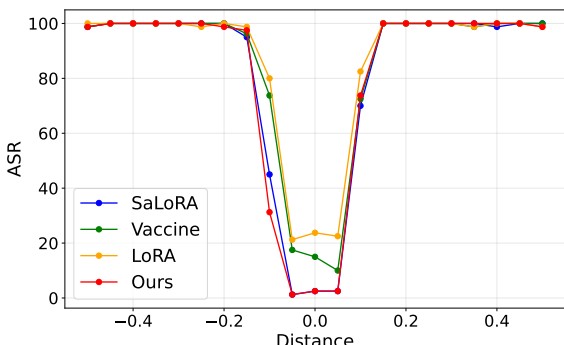

Figure 4: Safety landscape visualization showing Attack Success Rate (ASR) across parameter perturbations for different methods on Qwen 2.5 7B. Our approach maintains a significantly wider and deeper safety basin, with near 0% ASR at the origin and slower degradation with distance.

### 4.3.2 Parameter Perturbation Stability

We further evaluate the robustness of alignment restoration methods under parameter perturbations by analyzing the *safety basin* Peng et al. (2024a), which refers to the region in parameter space where the model continues to behave safely despite small changes.

**Experimental Setup** We test the Qwen 2.5 7B Instruct model fine-tuned with four methods: vanilla LoRA (as the baseline), and three safety alignment techniques: Vaccine Huang et al. (2024), SaLoRA Li et al. (2025a), and our curvature-aware approach. For each model, we apply parameter perturbations along randomly sampled directions, varying the perturbation magnitude within the range $[-0.5, 0.5]$.

We compare the **attack success rate (ASR)** at each perturbation level and compute the VISAGE score Peng et al. (2024a), which measures the average safety margin across all directions. A higher VISAGE score indicates that the model remains safe under a wider range of parameter variations.

**Results Analysis** As shown in Table 3 and Figure 4, our curvature-aware method achieves the highest VISAGE score (56.1), substantially outperforming SaLoRA (32.1), Vaccine (28.8), and LoRA (21.1). The safety landscape visualization confirms that our method maintains a broader and deeper safety basin, with nearly zero ASR at the origin and slower degradation as perturbation magnitude increases. These results indicate that our method produces more resilient safety alignment, offering stronger robustness to parameter noise and adaptation.

## 5 Conclusion

We present a curvature-aware alignment restoration framework that addresses the challenge of safety degradation in fine-tuned LLMs. Our approach builds on the empirical observation that the loss landscape associated with harmful content remains structurally preserved after task-specific

---

[1]Details of the non-refusal token construction are provided in Appendix B.4.

fine-tuning. Leveraging this geometric insight, we apply influence functions and second-order optimization to selectively increase loss on harmful inputs while maintaining task performance. Extensive evaluations across multiple model families and adversarial settings demonstrate that our method consistently reduces harmful responses while preserving few-shot generalization and utility on downstream tasks.

**Discussion**   While our work focuses on safety alignment restoration, the proposed curvature-aware framework is mathematically general and may extend to other scenarios involving conflicting objectives. The key insight enabling our approach is the geometric separation between task-relevant regions in the loss landscape. For safety, we empirically observe that harmful content occupies a structurally distinct region that remains preserved after fine-tuning. This property may not universally hold for arbitrary task pairs, when two objectives share substantial parameter dependencies, small curvature-constrained updates may be insufficient to optimize one without degrading the other. Investigating which objective pairs exhibit favorable geometric separation and developing metrics to predict this property a priori represents a promising direction for future research.

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

## A   Related Work

### A.1   Safety and Robustness in Large Language Models

**Safety Alignment in Large Language Models**   Ensuring the safety of large language models (LLMs) has become a central research challenge as their deployment expands into high-stakes domains. Models pretrained on vast internet corpora often internalize harmful behaviors, prompting the development of post-training alignment methods such as Reinforcement Learning from Human Feedback (RLHF) Ouyang et al. (2022); Dai et al. (2023) and supervised instruction tuning Bai et al. (2023); Zhang et al. (2023); Zhou et al. (2023). Despite their effectiveness, these safety mechanisms

remain fragile, with studies showing that fine-tuning aligned models on downstream tasks can lead to significant safety degradation Qi et al. (2023a;b). Concurrently, *parameter-efficient fine-tuning* (PEFT) techniques have emerged to adapt large models with minimal updates. Low-Rank Adaptation (LoRA) Hu et al. (2021) has become particularly popular by constraining updates to low-rank matrices applied to the model's weight matrices, significantly reducing trainable parameters while maintaining performance. Building on LoRA's efficiency, several *safety-preserving fine-tuning* approaches have been developed to address safety degradation. Vaccine Huang et al. (2024) introduces adversarial perturbations during training to immunize models against unsafe queries. SafeLoRA Hsu et al. (2024) extends LoRA by projecting weight updates onto an alignment subspace defined by the difference between aligned and unaligned model weights. Similarly, SaLoRA Li et al. (2025a) preserves safety during LoRA fine-tuning by introducing a fixed safety module that projects new features to a subspace orthogonal to original safety features. However, these approaches typically either compromise task performance or rely on heuristic projections without geometric insights. Recent findings suggest that safety-relevant behaviors occupy distinct, resilient regions in the loss landscape Peng et al. (2024b), indicating that geometric properties of the parameter space could enable more robust alignment preservation Li et al. (2025b); Arditi et al. (2024). Our work builds on these geometric insights by employing influence functions and curvature-aware optimization to restore safety alignment without sacrificing task performance. Unlike previous approaches that use heuristic constraints, our method directly leverages the preserved structure of the loss landscape to navigate toward parameter configurations that enhance safety while maintaining model capabilities.

## A.2 Unlearning and Parameter Space Geometry

When harmful behaviors emerge in LLMs following fine-tuning, *machine unlearning* offers a principled framework to selectively remove them Huu-Tien et al. (2024); Li et al. (2024); Liu et al. (2024). Influence function-based unlearning Koh & Liang (2017); Chen et al. (2023); Yuan et al. (2024) estimates the gradient direction that increases the loss on undesired examples while minimally impacting desired behaviors, effectively reversing their influence in parameter space Liu et al. (2025); Barez et al. (2025). Other approaches such as SISA Bourtoule et al. (2021b) or trust-region unlearning Golatkar et al. (2020) offer certified deletion by retraining from strategically partitioned checkpoints. However, these methods often incur high computational costs or suffer from degraded generalization. In parallel, *curvature-aware optimization* techniques have been explored to control model drift during fine-tuning. Elastic Weight Consolidation Kirkpatrick et al. (2017) and similar continual learning strategies use curvature estimates (e.g., Fisher information) to constrain updates in directions that preserve previously acquired capabilities. Trust-region policy optimization Schulman et al. (2015) and natural gradient methods Amari (1998) apply second-order constraints to keep parameter updates within functionally safe neighborhoods. Our method unifies these perspectives by framing safety restoration as a second-order constrained optimization problem over the loss landscape. We employ influence functions and L-BFGS-based curvature estimation to direct updates that increase loss on harmful content while staying within a trust region defined by the retain set, enabling scalable and stable safety restoration in fine-tuned LLMs.

## B    Detailed Implementation

### B.1    Curvature-Aware L-BFGS Construction

**Data Partitioning.**    To avoid overlap between curvature estimation and influence-based safety restoration, we partition the HEx-PHI dataset Qi et al. (2023b), which contains 330 adversarially constructed harmful prompts. For L-BFGS curvature pair construction, we use a total of 256 examples (64 examples from each of four batches) selected from HEx-PHI as part of the mixed curvature set $D_{\text{forget}}^{\text{curv}}$. Separately, to compute the forget loss $\mathcal{L}_{\text{forget}}$, we reserve 50 held-out examples from the remaining HEx-PHI data (named as $\mathcal{D}_{\text{forget}}$). These examples are not used during curvature approximation and are exclusively employed to evaluate or guide updates that suppress harmful generations. This partitioning ensures clean separation between curvature modeling and influence-based optimization targets.

To approximate the inverse Hessian $H_{\text{retain}}^{-1}$ in Equation 5, we construct a low-rank L-BFGS history over LoRA parameters using a carefully designed curvature buffer. This buffer integrates information from three strategically selected disjoint datasets: a subset of the forget set $\mathcal{D}_{\text{forget}}^{\text{curv}}$ (HEx-PHI dataset) and two distinct subsets of the retain set $\mathcal{D}_{\text{retain}}^{(1)}, \mathcal{D}_{\text{retain}}^{(2)}$ (derived from the fine-tuning dataset). This multi-dataset approach ensures the captured curvature spans both safety-critical and task-aligned directions in parameter space.

Each L-BFGS pair $(s_t, y_t)$ is computed via gradient accumulation over batches of 64 examples. Our empirical analysis reveals that just 10 high-quality pairs sufficiently approximate the local curvature structure for effective influence updates. We allocate these pairs approximately equally across the three datasets, requiring a minimum of 192 examples per set. To enhance curvature diversity, we employ varying learning rates (0.001, 0.002, 0.005) across optimization steps. A trust region $\delta_t$ constrains update magnitudes by scaling steps to a bounded norm, while a reduction ratio $\rho_t$ determines step acceptance and dynamically adjusts $\delta_t$.

To ensure robust and numerically stable curvature estimation, we implement several filtering mechanisms: **(1)** rejecting curvature pairs with insufficient curvature ($\langle s_t, y_t \rangle < 10^{-6}$), **(2)** normalizing $s_t, y_t$ vectors to unit norm before storage, **(3)** applying adaptive damping when negative curvature is encountered, and **(4)** excluding pairs with degenerate step or gradient norms. These safeguards collectively prevent ill-conditioning in the inverse Hessian approximation.

In practice, we recompute curvature pairs at the beginning of each safety restoration iteration. Our experiments demonstrate that just three such iterations suffice for effective alignment restoration across all evaluated model architectures, and this 3-step procedure is consistently employed throughout our experimental validation.

---

**Algorithm 1** Curvature-Aware L-BFGS History Construction

---

1: **Input:** Model $f_\theta$, datasets $\mathcal{D}_{\text{forget}}^{\text{curv}}, \mathcal{D}_{\text{retain}}^{(1,2)}$, LoRA parameters $\theta_{\text{LoRA}}$
2: Initialize empty history lists: $\mathcal{S}, \mathcal{Y}$
3: Set initial trust radius $\delta = 0.05$
4: **for** $t = 1$ to $T$ **do**
5:     Sample batch $B_t$ from one of the datasets (round-robin)
6:     Compute initial loss $\mathcal{L}_{\text{init}}$ and gradients $g_t$
7:     Propose step $d_t = -g_t$ and rescale to $\|d_t\| \leq \delta$
8:     Save $\theta_t$, apply step to get $\theta_{t+1}$
9:     Compute final loss $\mathcal{L}_{\text{final}}$ and gradients $g_{t+1}$
10:     Compute actual and predicted reduction, ratio $\rho_t$
11:     **if** $\rho_t < 0.25$ **then**
12:         Shrink trust radius: $\delta \leftarrow 0.5\delta$
13:         Revert to $\theta_t$
14:         **continue**
15:     **else if** $\rho_t > 0.75$ **then**
16:         Expand trust radius: $\delta \leftarrow 1.5\delta$
17:     **end if**
18:     Compute $s_t = \theta_{t+1} - \theta_t$, $y_t = g_{t+1} - g_t$
19:     **if** $\langle s_t, y_t \rangle > \epsilon$ **then**
20:         Normalize $s_t$, $y_t$, add to $\mathcal{S}, \mathcal{Y}$
21:     **end if**
22: **end for**
23: **return** $\mathcal{S}, \mathcal{Y}$

---

## B.2 Influence Update Mechanism

To prevent overcorrection and preserve generalization capabilities of the model during alignment restoration, we apply L2 regularization to the influence-based update direction $\Delta\theta$. At each iteration, the update to the LoRA parameters is computed as:

$$\theta_{\text{new}} = \theta_{\text{tuned}} + \eta \cdot \Delta\theta - \lambda \cdot \theta,$$

where:

- $\eta$ is the update scale (determined by a fixed multiplier or small grid search),

- $\Delta\theta$ is the L-BFGS-projected gradient direction (from Appendix B.1),

- $\lambda$ is the L2 regularization weight, progressively annealed across iterations (e.g., $\lambda \leftarrow 0.95 \cdot \lambda$).

**Unlearning Objective** The harmful gradient $\nabla\mathcal{L}_{\text{forget}}$ is obtained by evaluating the model on the forget set using a cross-entropy loss:

$$\mathcal{L}_{\text{forget}} = \text{CE}(\hat{y}, y)$$

---

**Algorithm 2** Safety Restoration via Influence Update

---

1: **Input:** LoRA parameters $\theta$, L-BFGS history $(\mathcal{S}, \mathcal{Y})$, forget dataset $\mathcal{D}_{\text{forget}}$, step size $\eta$, L2 weight $\lambda$
2: Initialize accumulated gradient $g \leftarrow 0$
3: **for** each batch $B$ in $\mathcal{D}_{\text{forget}}$ **do**
4:     Compute loss $\mathcal{L}_{\text{forget}} = \text{CE}(f_\theta(B))$
5:     Compute gradient $\nabla \mathcal{L}_{\text{forget}}$ and accumulate into $g$
6: **end for**
7: Project $g$ through inverse Hessian: $\Delta\theta \leftarrow -H^{-1}g$ using L-BFGS (see Appendix B.1)
8: Normalize $\Delta\theta \leftarrow \Delta\theta/\|\Delta\theta\|$
9: **for** each parameter $\theta_i$ in LoRA:
10:     Extract corresponding slice $\Delta\theta_i$
11:     Compute L2-regularized update:

$$\theta_i \leftarrow \theta_i + \eta \cdot \Delta\theta_i - \lambda \cdot \theta_i$$

12: **return** Updated parameters $\theta$

---

### B.3 Loss Landscape Visualization Implementation

This section provides a detailed description of our methodology for visualizing the loss landscapes of language models before and after fine-tuning.

**Gradient-Informed Direction Generation**   Unlike conventional approaches that use random directions in parameter space, we generate perturbation directions informed by gradients computed on the model's loss function. For computational tractability, we focus only on attention and MLP layers, which most strongly influence model behavior. For each perturbation direction $\mathbf{d}_i$, we calculate:

$$\mathbf{d}_i = \text{RandomScale}(\nabla_\theta \mathcal{L}(\theta)) \tag{6}$$

where $\nabla_\theta \mathcal{L}(\theta)$ represents accumulated gradients from a fixed set of validation samples, and RandomScale($\cdot$) applies random scaling factors to different parameters using a direction-specific random seed. We generate two perturbation directions $\mathbf{d}_1$ and $\mathbf{d}_2$ using different random seeds (1000 and 2000), which affects the scaling factors applied to the gradients. Due to the high dimensionality of the parameter space, these two directions are approximately orthogonal with high probability.

**Grid Construction and Evaluation**   To visualize the loss landscape, we construct a 2D grid in parameter space by varying perturbation magnitudes along these two directions:

$$\theta_{i,j} = \theta_{\text{original}} + \lambda_i \cdot \mathbf{d}_1 + \lambda_j \cdot \mathbf{d}_2 \tag{7}$$

where $\lambda_i, \lambda_j \in [-\alpha, \alpha]$ are scalar coefficients with $\alpha = 0.01$. We construct a $20 \times 20$ grid by uniformly sampling $\lambda$ values. For each grid point $\theta_{i,j}$, we compute the model's loss on both harmful and benign datasets, creating separate loss landscapes for each model state.

**Memory-Optimized Implementation**   Large language models present significant memory challenges for loss landscape visualization. To address this, we implement several optimizations: row-by-row processing to compute one grid row at a time; parameter subsetting that applies perturbations only to attention and MLP layers; gradient accumulation over small batches; and bfloat16 precision for all computations. These techniques allow us to visualize loss landscapes of multi-billion parameter models without excessive memory requirements.

**Structural Difference Quantification**   To quantify the structural similarity between base and fine-tuned model loss landscapes, we define a correlation-based metric:

$$\text{StructDiff} = (1 - |\text{corr}(\nabla^2 \mathcal{L}_{\text{base}}, \nabla^2 \mathcal{L}_{\text{tuned}})|) \times 100\% \tag{8}$$

where $\nabla^2 \mathcal{L}$ is the Laplacian of the loss landscape, approximated using finite differences on our computed grid. This metric captures differences in curvature patterns rather than absolute loss values, providing a more reliable measure of structural preservation. The figure visualizes a visualization on LLama-3-8B, under LoRA fine-tuning (where hyperparameters are used the same as in the main experiments), which further confirms the model preserver structure on harmful data compared to the benign ones.

**Cross-Sectional Analysis**   To provide additional insight into the loss landscape structure, we extract cross-sectional views along each perturbation direction at the origin point. Figure 6 shows these cross-sections across our three evaluation datasets. For Dolly and Alpaca datasets (left and middle columns), we observe significant structural divergence between base and fine-tuned models. The fine-tuned model consistently exhibits lower loss values in negative direction regions, reflecting optimization for task-specific objectives. The intersection points where the curves cross represent transition zones in parameter space where model behaviors begin to diverge more dramatically.

On the other hand, for harmful content (right column), the base and fine-tuned model loss curves remain remarkably parallel with nearly identical structural features. Both models show similar sharp increases in loss (forming "cliff" patterns) at similar positions along both directions. This preserved geometric correspondence provides quantitative evidence for our hypothesis that safety-relevant regions in the loss landscape maintain their structural integrity during fine-tuning. These cross-sectional visualizations complement our 3D surface plots and structural difference metrics, providing a more granular view of how loss landscapes change along specific directions of interest.

### B.4   Prefill Attack Construction and Non-Refusal Token Injection

To simulate prefilling attacks, we adopt the setup introduced in AdvBench Zou et al. (2023), which provides a collection of adversarial goal-target pairs designed to bypass refusal mechanisms in safety-aligned language models. Each goal represents a harmful instruction, and the corresponding target is a benign-looking prefix that avoids immediate refusal while steering the model toward unsafe completions. In our setup, we construct the prefilled input by first applying a prompt template to each goal, then appending the associated target prefix directly to the end of the prompt. The resulting input is passed to the model, forcing it to generate from a context that includes several non-refusal tokens. We use a fixed number of prefix tokens (e.g., the first 4 tokens from each target) to

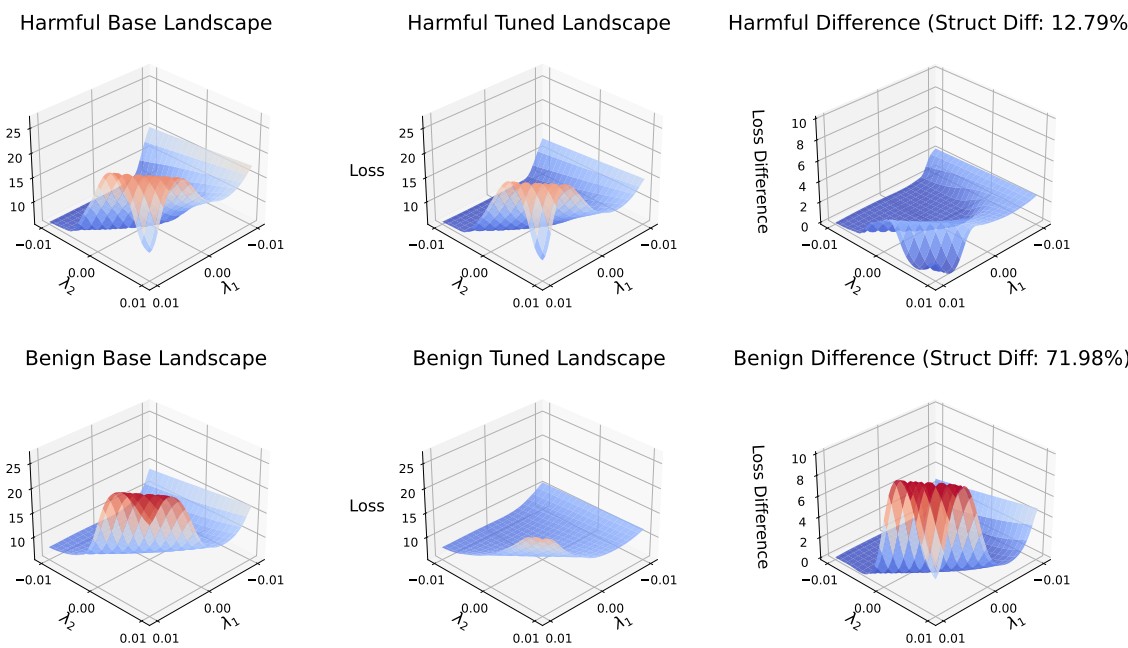

Figure 5: 3D loss landscape visualization for LLaMA-3 8B with LoRA fine-tuning using gradient-informed direction projection. Top row: harmful content (HEx-PHI); bottom row: general data (Alpaca). LoRA fine-tuning preserves the loss landscape structure for harmful content (12.79% structural difference) while substantially altering general data landscapes (71.98% structural difference), demonstrating that parameter-efficient methods similarly maintain safety-relevant geometric features.

ensure consistent perturbation across examples. This approach effectively bypasses shallow safety filters by shifting the harmful intent away from the beginning of the prompt, thereby exposing vulnerabilities in the model's alignment mechanisms.

## C  Ablation Study

### C.1  Hyperparameter Sensitivity

We evaluate our method's sensitivity to key hyperparameters on LLaMA-3.1 8B: retain loss threshold $\epsilon$, trust region radius $\delta$, and forget set size (Table 4).

Our method demonstrates robustness across practical ranges: $\epsilon \in [0.1, 0.4]$ maintains HRR $\leq 5.5\%$ with stable utility, $\delta \in [0.05, 0.2]$ shows consistent performance, and even with 50 samples, effective restoration is achieved. Default settings ($\epsilon = 0.1$, $\delta = 0.1$, 50 samples) generalize well across all tested models.

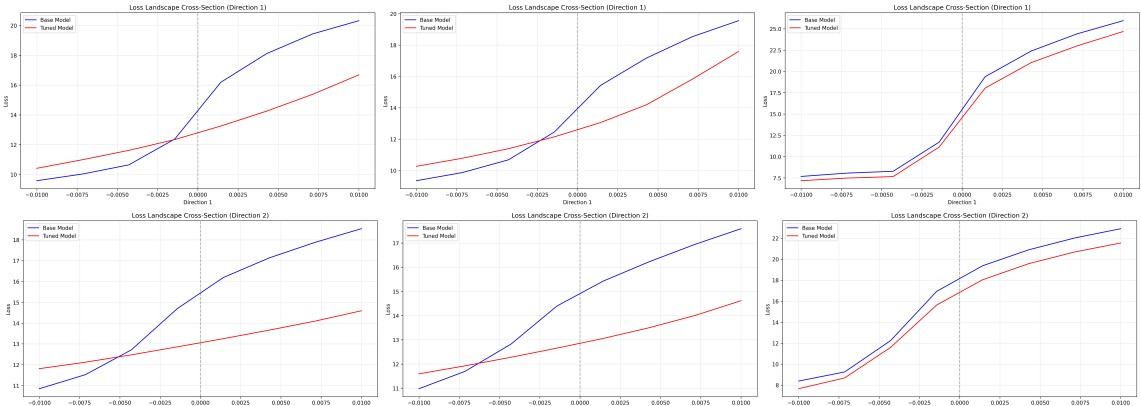

Figure 6: Loss landscape cross-sections along two perturbation directions for base (blue) and fine-tuned (red) models utilizing Qwen 2.7 7B Instruct across three datasets: Dolly (left), Alpaca (middle), and HEx-PHI harmful content (right). While task-specific and general datasets show significant divergence between models, harmful content exhibits remarkable structural similarity with preserved curvature characteristics, particularly near the origin (0,0).

Table 4: Hyperparameter sensitivity analysis showing robust performance across practical parameter ranges.

| $\epsilon$ | HRR$\downarrow$ | ARC-C | BoolQ | $\delta$ | HRR$\downarrow$ |
|---|---|---|---|---|---|
| 0.1 | 5.5 | 52.1 | 84.0 | 0.01 | 7.5 |
| 0.2 | 3.0 | 52.1 | 84.0 | 0.05 | 3.0 |
| 0.3 | 3.0 | 52.1 | 84.2 | 0.10 | 3.0 |
| 0.4 | 2.0 | 52.4 | 84.2 | 0.20 | 3.5 |

| Set Size | HRR$\downarrow$ | HellaSwag | WinoGrande |
|---|---|---|---|
| 50 | 3.0 | 59.1 | 74.3 |
| 100 | 3.0 | 59.1 | 73.8 |
| 150 | 4.0 | 59.1 | 73.8 |
| 200 | 3.5 | 59.1 | 74.4 |

## C.2 Safety Restoration Under Full Fine-tuning

To verify that our curvature-aware restoration method generalizes beyond parameter-efficient fine-tuning (LoRA) to full fine-tuning scenarios, we applied our approach to fully fine-tuned models. Table 5 presents the safety restoration results alongside task performance metrics.

Our method achieves substantial reductions in harmful response rates: 57% for LLaMA-2 7B (from 52.0% to 22.5%) and 75% for LLaMA-3.1 8B (from 42.0% to 10.5%). Critically, these safety improvements are achieved while maintaining or slightly improving task performance across all benchmarks. For LLaMA-3.1 8B, we observe notable improvements in TruthfulQA (+8.5 points)

Table 5: Safety restoration results for full fine-tuning. Our curvature-aware method significantly reduces harmful response rates (HRR) while maintaining or improving performance on utility benchmarks (TruthfulQA, ToxiGen, ARC-C, BoolQ).

| Model | Method | HRR ↓ | TruthfulQA | ToxiGen | ARC-C | BoolQ |
|---|---|---|---|---|---|---|
| LLaMA-2 7B | Original (full FT) | 52.0 | 33.2 | 56.0 | 21.4 | 62.3 |
| | + Ours | **22.5** | 34.5 | 56.8 | 20.7 | 62.1 |
| LLaMA-3.1 8B | Original (full FT) | 42.0 | 36.9 | 42.8 | 51.6 | 83.1 |
| | + Ours | **10.5** | 45.4 | 46.9 | 53.3 | 84.2 |

Table 6: In-context learning performance on six commonsense reasoning tasks using Llama-2 7B Chat. Results show 5-shot accuracy percentages with improvements over zero-shot in parentheses. Our curvature-aware method achieves the highest few-shot learning gains on five of six tasks, demonstrating that safety restoration preserves and enhances the model's ability to leverage examples. Bold indicates best absolute performance, while underlines highlight the largest zero-to-five-shot improvements.

| Methods | ARC-Easy | BoolQ | PIQA | HellaSwag | ARC-Challenge | WinoGrande |
|---|---|---|---|---|---|---|
| LoRA | 78.2 (+1.0) | 81.5 (+4.6) | 77.6 (-0.1) | 55.6 (+0.0) | 46.2 (+1.8) | **72.5** (+3.5) |
| Vaccine | 77.2 (+1.7) | **82.4** (+5.0) | 77.1 (-0.8) | 54.6 (+0.3) | 44.3 (+1.7) | 71.1 (+3.6) |
| SaLoRA | 79.4 (+3.0) | 82.3 (+3.5) | 78.0 (-0.2) | 57.3 (+0.3) | 48.3 (+2.4) | **72.5** (+3.8) |
| Safe LoRA | 78.9 (+2.6) | 80.7 (+2.2) | 77.8 (-0.7) | 56.5 (+0.0) | 46.8 (+1.2) | 72.2 (+4.1) |
| **Ours** | **79.8** (+4.4) | 82.2 (+2.5) | **78.2** (+1.3) | **58.7** (+0.9) | **49.7** (+5.0) | 72.2 (+4.4) |

and ToxiGen (+4.1 points), demonstrating that safety restoration does not necessitate utility trade-offs.

These results confirm that the preserved loss landscape geometry identified in Section 2.1 that harmful content maintains high correlation (0.85–0.99) even under full fine-tuning. This enables effective curvature-aware restoration regardless of whether parameter-efficient or full fine-tuning is employed. The slightly higher HRR compared to LoRA-based restoration (Table 2) aligns with the observation that full fine-tuning exhibits somewhat lower correlation (e.g., 0.852 vs 0.992 for LLaMA-2 7B in Table 1), yet restoration remains highly effective.

### C.3  In-Context Learning Performance

We assess in-context learning capability via a few-shot evaluation to determine if alignment restoration preserves the model's adaptability. We measure how different safety restoration methods affect Llama-2 7B's few-shot learning performance across six commonsense reasoning benchmarks. For each task, we compare zero-shot performance with 5-shot performance, where five task examples are included in the prompt before the test instance, allowing the model to perform in-context learning. The improvement from zero-shot to 5-shot performance reflects the model's ability to leverage examples for rapid adaptation, a fundamental capability that should remain intact after safety restoration.

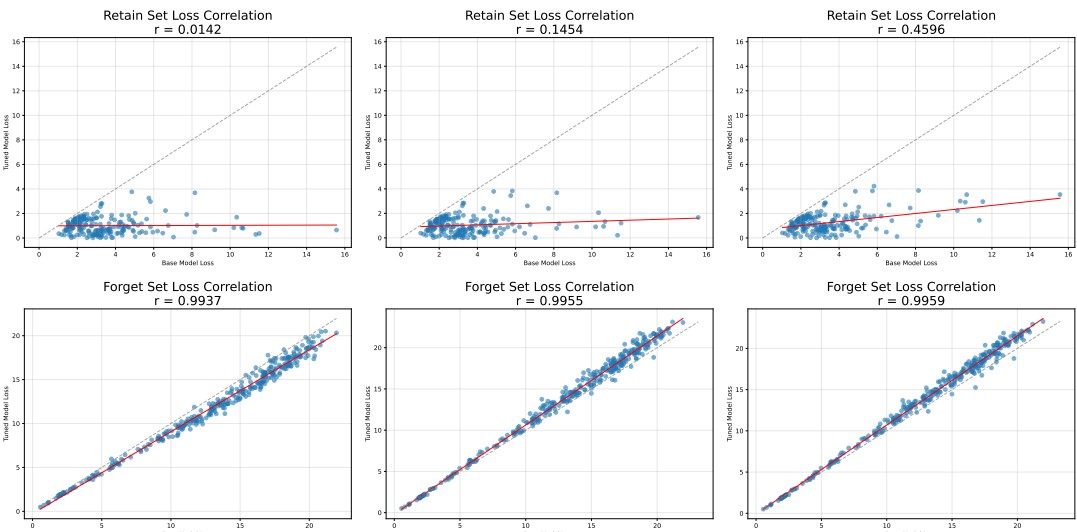

Figure 7: Correlation analysis during restoration. Top row: Dolly (test set) correlation improves from $r = 0.014$ to $r = 0.456$, showing functional recovery. Bottom row: Forget set correlation stabilizes at $r = 0.996$, demonstrating realignment with base model behavior in harmful regions.

In Table 6, our method demonstrates the highest few-shot learning gains on five of six tasks. On ARC-Easy, our approach achieves a substantial $+4.4\%$ improvement over zero-shot, significantly outperforming all baselines, including SaLoRA ($+3.0\%$) and Safe LoRA ($+2.6\%$). This pattern continues across other tasks, most notably on ARC-Challenge, where our method achieves a remarkable $+5.0\%$ improvement, more than double that of SaLoRA ($+2.4\%$).

Notably, our method shows a $+1.3\%$ improvement on PIQA, while all other methods demonstrate minimal or negative transfer. This suggests our curvature-aware approach better preserves the model's commonsense physical reasoning capabilities, which are particularly sensitive to parameter modifications.

### C.4  Recovery of Base Model Behavior

A key result of our alignment restoration approach is its ability to recover the original base model's safety behavior patterns. To verify this property, we analyze the relationship between the restored model and the base model throughout the recovery process on Qwen 2.5 7B Instruct model. Figure 7 illustrates Pearson correlation coefficients between the base model and the restored model on a held-out evaluation set (Dolly) under increasing restoration steps. We report correlations on both the retain (Dolly) set and forget (harmful) set, computed between per-example loss values as a proxy for functional alignment.

Initially, the fine-tuned (unsafe) model exhibits near-zero correlation with the base model on the retain set (e.g., $r = 0.014$), indicating severe deviation. As alignment restoration progresses, the correlation increases steadily (e.g., $r = 0.145$, then $r = 0.460$), reflecting functional recovery. On

the forget set, we observe near-perfect preservation of the base model's loss ranking by the third step ($r = 0.996$), suggesting that the restored model re-aligns closely with the base behavior in harmful regions. These results support the hypothesis that the safety properties of the base model remain geometrically accessible even after fine-tuning, and that our method effectively re-navigates the loss landscape to recover them.

### C.5   Connection to Machine Unlearning

Our safety restoration approach shares similarities with machine unlearning—both involve modifying model behavior on specific data subsets—but differ in crucial ways regarding objectives, mechanisms, and constraints.

Machine unlearning aims to remove knowledge or capabilities from a model, making it behave as if certain training data were never seen (Yao et al., 2024; Bourtoule et al., 2021b; Maini et al., 2024). This typically requires the model to "forget" how to produce specific outputs. In contrast, our safety restoration leverages the empirical observation that alignment mechanisms remain structurally preserved after fine-tuning (Section 2.1), with safety behaviors merely suppressed rather than erased. Rather than eliminating capabilities, we aim to restore appropriate refusal responses that already exist in the model's loss landscape.

This conceptual difference leads to distinct technical challenges. Previous work  (Yuan et al., 2024) categorize unlearning methods into *untargeted* approaches that maximize loss on forget data, and *targeted* approaches that train models to produce specific template responses. Untargeted methods face an output quality problem: higher loss can result from random token sequences or incoherent text rather than the coherent refusals desired for safety. Targeted methods encounter a different issue, because forget and retain inputs are distributionally similar, increasing the probability of refusal templates on forget examples also increases refusal probability on retain examples, producing what they term "excessive ignorance" where models incorrectly refuse benign requests.

Additionally, effective unlearning typically requires substantial parameter updates to meaningfully alter output distributions and remove memorized patterns. Our method exploits the preserved geometric structure to achieve safety restoration through precise, localized updates that avoid catastrophic forgetting on task-relevant knowledge.

To empirically assess whether standard unlearning techniques can effectively restore safety, we compare our curvature-aware method against three representative baselines on LLaMA-3.1 8B:

- **Gradient Ascent:** $\theta \leftarrow \theta + \eta \nabla_\theta \mathcal{L}_{\text{forget}}$

- **GradDiff:** $\theta \leftarrow \theta - \eta(\nabla_\theta \mathcal{L}_{\text{retain}} - \alpha \nabla_\theta \mathcal{L}_{\text{forget}})$, $\alpha \in \{0.5, 1.0, 2.0\}$

All methods use identical data and comparable parameter update budgets (controlled via $\epsilon$).

Table 7 reveals a critical disconnect between forget loss and safety outcomes. GradDiff with $\alpha = 0.5$ achieves the highest forget loss (44.0) yet provides minimal safety improvement (HRR 22.0%). More strikingly, our method achieves the smallest forget loss compared to other methods. This confirms that maximizing loss alone does not guarantee appropriate refusal behavior—models may generate incoherent outputs rather than helpful refusals. Our curvature-aware approach navigates toward

Table 7: Comparison with unlearning methods on LLaMA-3.1 8B under similar update budgets. Forget Loss measures the model's cross-entropy loss on harmful content (higher indicates the model assigns lower probability to harmful responses).

| Method | Forget Loss ↑ | HRR ↓ | TruthfulQA | ToxiGen | ARC-C | BoolQ |
|---|---|---|---|---|---|---|
| Fine-tuned | 8.8 | 25.5 | 43.6 | 46.0 | 51.8 | 84.0 |
| Gradient Ascent | 25.9 | 24.5 | 34.8 | 43.6 | 30.5 | 73.8 |
| GradDiff ($\alpha$=0.5) | 44.0 | 22.0 | 34.5 | 55.7 | 20.2 | 43.6 |
| GradDiff ($\alpha$=1.0) | 19.0 | 27.0 | 36.5 | 43.3 | 18.8 | 55.8 |
| GradDiff ($\alpha$=2.0) | 17.9 | 23.8 | 35.5 | 43.0 | 47.5 | 82.1 |
| **Ours** | **15.5** | **3.0** | **45.4** | **46.9** | **53.3** | **84.2** |

semantically meaningful refusals while maintaining utility, whereas first-order methods degrade performance substantially (e.g., ARC-C drops to 18.8–30.5 vs our 53.3).

## C.6  Comparison with First-Order Methods

First-order optimization methods dominate machine unlearning approaches due to their computational efficiency. However, these methods struggle with the complex non-convex landscapes characteristic of fine-tuned LLMs. Our curvature-aware approach fundamentally improves upon first-order methods by incorporating second-order information about the loss landscape's geometry. Figures 8 and 9 visualize the optimization trajectories of our curvature-aware method versus a representative first-order approach at different learning rates (relate to first-order methods, we choose gradient ascent for simplicity). We project the high-dimensional parameter space into a 2D representation using Principal Component Analysis (PCA) on parameter updates during optimization. The contour plots represent the combined objective landscape, where higher values (yellow regions) indicate better safety restoration while preserving task performance.

At a conservative learning rate (0.01, Figure 8), the first-order method (blue trajectory) exhibits inefficient navigation, following a suboptimal path that initially makes progress but then traverses through lower-value regions. In contrast, our curvature-aware approach (red trajectory) identifies and follows a more direct path toward the high-value region, demonstrating superior awareness of the landscape's geometry. At a higher learning rate (0.05, Figure 9), the limitations of first-order methods become even more pronounced. The blue trajectory exhibits dramatic oscillations and instability, making large, erratic movements through parameter space. Our curvature-aware method maintains remarkable stability even at this higher learning rate, following an almost perfectly straight path that steadily progresses through increasingly favorable regions of the objective landscape.

## C.7  Quantitative Analysis of Loss Landscape Preservation

To complement the visual analysis in Figure 2, we provide quantitative statistics of loss differences across the perturbation grid. Table 8 reports percentile distributions of $\mathcal{L}_{\text{tuned}} - \mathcal{L}_{\text{base}}$ for both harmful and benign content.

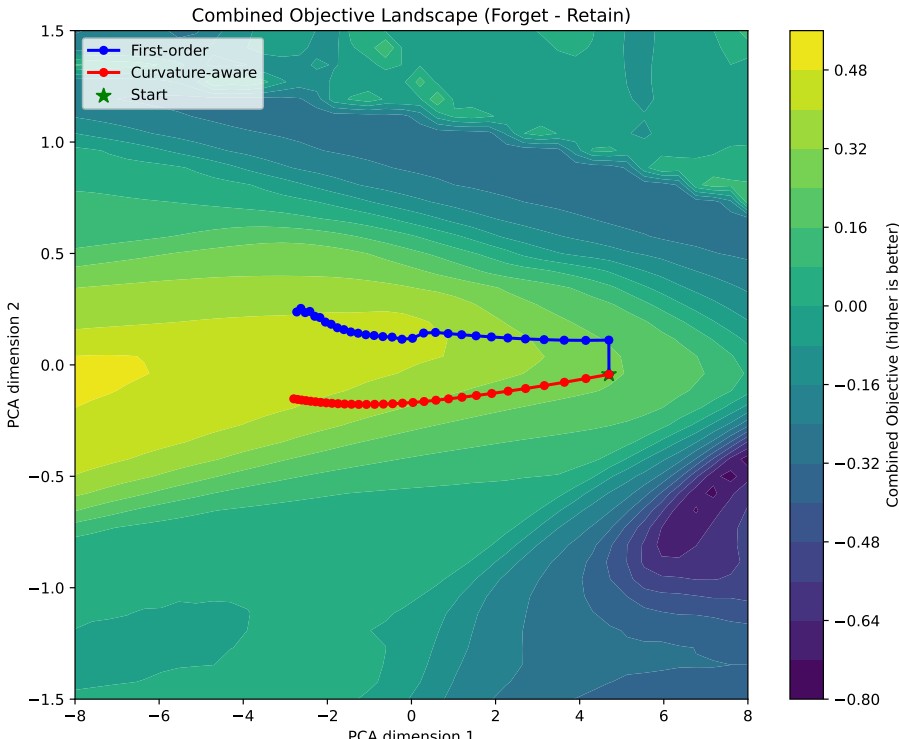

Figure 8: Parameter space navigation comparison between first-order (blue) and curvature-aware (red) methods at a conservative learning rate (0.01), projected onto the first two principal components. The contour plot shows the combined objective landscape (forget loss minus retain loss), where higher values (yellow) represent more effective safety restoration while preserving task performance. Our curvature-aware approach follows a more direct path through higher-value regions, demonstrating superior landscape navigation. Both methods start from the same fine-tuned model parameters (green star).

Harmful content exhibits tighter distribution (Std=0.387) than benign content (Std=0.810). Across all percentiles, absolute differences $|\mathcal{L}_{\text{tuned}} - \mathcal{L}_{\text{base}}|$ are systematically smaller for harmful content, confirming more consistent loss structure preservation. Note that the "Struct Diff" metric in Figure 2 measures curvature correlation rather than absolute loss magnitudes—two landscapes can have similar loss ranges but different geometric structure. Figure 6 provides cross-sectional views showing parallel curves for harmful content versus divergent curves for benign content.

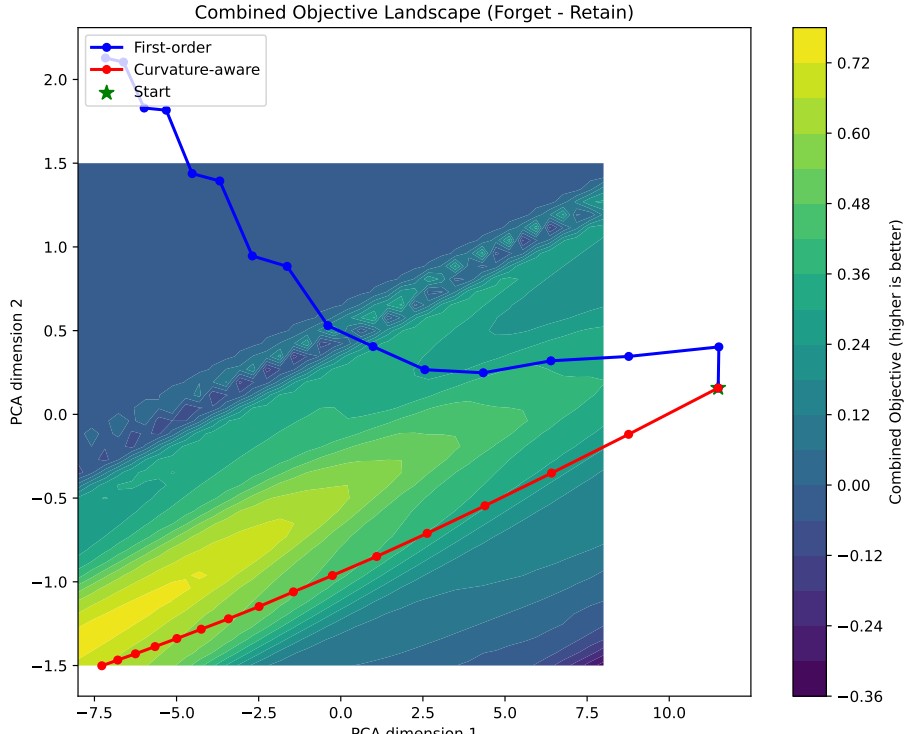

Figure 9: Parameter space navigation comparison at a higher learning rate (0.05). The first-order method (blue) exhibits extreme oscillation and instability, making large erratic movements and repeatedly venturing into negative-value regions (purple/dark blue). In contrast, our curvature-aware method (red) demonstrates remarkable stability, following an almost perfectly straight path that steadily progresses through higher-value regions. This visualization highlights how curvature awareness provides robustness to hyperparameter choices and avoids wasteful exploration of the parameter space.

## C.8   Computational Cost

We evaluate computational efficiency by comparing runtime, memory usage, and inference overhead against baseline methods on LLaMA-2 7B using an NVIDIA H100 80GB GPU.

In Table 9, our method requires 18.5 minutes, which is comparable to SaLoRA (18.4 min) and faster than Vaccine (26.1 min). Higher memory consumption (48.9GB vs 22–34GB) stems from L-BFGS gradient history storage, but this is a one-time cost during restoration. Critically, we introduce no inference overhead unlike SaLoRA's +10% from additional safety modules, and require fewer harmful samples (50 vs 138). This represents a tradeoff between memory and accuracy: one-time

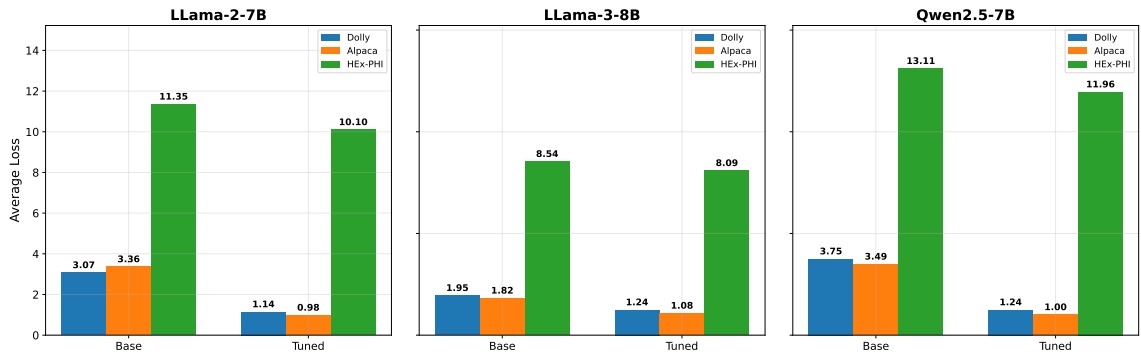

Figure 10: Average loss comparison across base and fine-tuned models for three datasets: Dolly (task-specific), Alpaca (general), and HEx-PHI (harmful). Across all three model families, harmful content consistently exhibits substantially higher loss (8.09-13.11) compared to benign content (0.98-3.75) in base models. Fine-tuning reduces loss on both task-specific and general content while simultaneously reducing the loss gap on harmful content.

Table 8: Percentile statistics of loss differences across perturbation grid (LLaMA-3 8B, Figure 2).

| Percentile | Harmful (Raw) | Benign (Raw) | Harmful (Abs) | Benign (Abs) |
|---|---|---|---|---|
| 10th | $-0.603$ | $-0.445$ | 0.102 | 0.203 |
| 20th | $-0.497$ | $-0.399$ | 0.177 | 0.279 |
| 30th | $-0.430$ | $-0.363$ | 0.215 | 0.327 |
| 40th | $-0.388$ | $-0.350$ | 0.247 | 0.348 |
| 50th | $-0.321$ | $-0.330$ | 0.325 | 0.360 |
| 60th | $-0.242$ | $-0.283$ | 0.388 | 0.380 |
| 70th | $-0.204$ | $-0.167$ | 0.430 | 0.433 |
| 80th | $-0.115$ | 0.248 | 0.497 | 0.582 |
| 90th | 0.099 | 0.672 | 0.603 | 0.878 |
| Mean | $-0.346$ | $-0.257$ | 0.389 | 0.538 |
| Std | 0.387 | 0.810 | 0.343 | 0.658 |

$2 \times$ memory overhead yields substantially better safety (3.0% vs 11.0% HRR) with no deployment cost.

Table 10 breaks down our restoration pipeline across architectures. L-BFGS construction takes 6–7 minutes; influence updates only 16–18 seconds. The curvature estimation dominates but remains tractable for modern LLMs.

Table 9: Comprehensive computational comparison. All methods measured on same hardware (H100 GPU) with LLaMA-2 7B.

| Method | Time (min) | Memory (GB) | Inference | Harmful Data |
|---|---|---|---|---|
| LoRA (baseline) | 14.3 | 22.0 | None | No |
| SafeLoRA | 15.0 | 22.0 | None | No |
| SaLoRA | 18.4 | 34.2 | +10% | Yes (138) |
| Vaccine | 26.1 | 24.6 | None | No |
| **Ours** | **18.5** | **48.9** | **None** | **Yes (50)** |

Table 10: Runtime breakdown per restoration iteration on H100 GPU with 50 harmful samples.

| Model | L-BFGS Construction (s) | Influence Update (s) |
|---|---|---|
| LLaMA-2 7B | 399 | 16 |
| LLaMA-3.1 8B | 433 | 18 |
| Qwen 2.5 7B | 409 | 17 |

