# OpenReview forum: "Curvature-Aware Safety  Restoration In LLMs Fine-Tuning"
_TMLR — Accepted by TMLR_

### Review · Reviewer_whRS · 2025-12-02

**Summary Of Contributions:**

This paper studies how to ensure safety alignment while fine-tuning LLMs. Motivated by some investigation on the geometric structure of the loss landscape, the authors propose a curvature-aware alignment restoration method. Their empirical results demonstrate effectiveness in discouraging unsafe outputs while preserving task-relevant performance.

**Additional Comments:**

1. Although this paper considers safety alignment v.s. preserving performance in target tasks, I'm curious whether this framework is also applicable for an arbitrary pair of two tasks with somewhat conflicts, where you want to optimize one of them and preserve the performance in the other.

**Audience:**

Yes

**Audience Explanation:**

It is an interesting and practically demanding topic to study how to fine-tuning model in target tasks with safety constraints. Contributions on understanding training behaviors or developing new algorithms are valuable.

**Claims And Evidence:**

Yes

**Claims Explanation:**

Most of the claims are supported by empirical observations.

I have some concerns about the results in Figure 2. Although there is a big difference in terms of "Struct Diff", according to the picture, the ranges of the loss difference seem not diverge much. I'm not sure if it is because I'm not good at interpretting the 3D picture, but it may be better to make it clear. For example, maybe report the 10,20,...,90 percentiles of the loss differences when varying $\lambda_1$ and $\lambda_2$ on grids.

**Requested Changes:**

1. See my remarks for Figure 2 above.

2. I'm not very familiar with the safety alignment literature, but I guess there is an obvious baseline missing: just optimize L_{retain} and L_{forget} together (e.g. $\min_\theta L_{retain}(\theta) - \alpha L_{forget}(\theta)$ for some appropriate hyper-parameter $\alpha$). I'm curious about whether the proposed method still has advantages compared with this simple baseline.

   Such a baseline can be interpreted as approximately optimizing the lagrangian of the constraint problem.

---

> ### Author Response · Authors · 2026-01-08
> **Response to Reviewer whRS**
>
> We thank Reviewer whRS for the positive evaluation and the insightful questions about theoretical connections. We address each concern below.
>
> ### Figure 2 Clarity
>
> > "The ranges of the loss difference seem not diverge much. Maybe report the 10,20,...,90 percentiles of the loss differences when varying λ₁ and λ₂ on grids."
>
> We appreciate this suggestion for improving clarity. We computed percentile statistics of the raw loss differences (L_tuned − L_base) across the perturbation grid, measured on the loss landscape of Figure 2 in the main paper:
>
> **Table R6: Percentile Statistics of Loss Differences (L_tuned − L_base)**
> | Percentile | Harmful (Raw) | Benign (Raw) | Harmful (Abs) | Benign (Abs) |
> |------------|---------------|--------------|---------------|--------------|
> | 10th | -0.603 | -0.445 | 0.102 | 0.203 |
> | 20th | -0.497 | -0.399 | 0.177 | 0.279 |
> | 30th | -0.430 | -0.363 | 0.215 | 0.327 |
> | 40th | -0.388 | -0.350 | 0.247 | 0.348 |
> | 50th | -0.321 | -0.330 | 0.325 | 0.360 |
> | 60th | -0.242 | -0.283 | 0.388 | 0.380 |
> | 70th | -0.204 | -0.167 | 0.430 | 0.433 |
> | 80th | -0.115 | 0.248 | 0.497 | 0.582 |
> | 90th | 0.099 | 0.672 | 0.603 | 0.878 |
> | Mean | -0.346 | -0.257 | 0.389 | 0.538 |
> | Std | 0.387 | 0.810 | 0.343 | 0.658 |
>
> The key observations are: (1) Harmful content shows a much tighter distribution (Std=0.387) than benign (Std=0.810), indicating more consistent loss structure preservation. (2) 90% of harmful samples maintain loss differences within a narrow range [-0.603, 0.099], while benign samples vary much more widely [-0.445, 0.672]. (3) The absolute differences |Harmful| are systematically smaller across all percentiles, confirming preserved loss landscape structure for safety-relevant content.
>
> We note that the "Struct Diff" metric in Figure 2 captures curvature correlation rather than absolute loss magnitude. Two landscapes can have similar loss ranges but different geometric structures (valley and peak locations). The cross-sectional plots in Appendix C.3 (Figure 6) provide complementary visualization showing parallel curves for harmful content versus divergent curves for benign content. We have added this percentile table and made a more prominent reference to Figure 6 in the revised manuscript, in Appendix D.5 (Quantitative Analysis of Loss Landscape Preservation).
>
> ### Missing Baseline: Simple Lagrangian
>
> > "There is an obvious baseline missing: just optimize L_retain and L_forget together (e.g., min_θ L_retain(θ) − αL_forget(θ))."
>
> We address this concern in **Common Concern 3** above (Table R4). The GradDiff baseline with various alpha values achieves higher forget loss compared to our method, but with dramatically worse safety, demonstrating that curvature-aware optimization provides benefits beyond simple gradient-based approaches.
>
> ### Generalizability to Arbitrary Task Pairs
>
> > "I'm curious whether this framework is also applicable for an arbitrary pair of two tasks with somewhat conflicts."
>
> We thank the reviewer for this insightful question. The mathematical framework (Equations 2–5) is indeed general and could apply to arbitrary conflicting task pairs. However, our empirical analysis reveals that safety restoration benefits from a particularly favorable geometric property: safety-relevant regions remain structurally preserved after fine-tuning.
>
> For arbitrary task pairs, the method's effectiveness would depend on whether similar geometric separation exists. When two tasks heavily share parameter dependencies, the curvature-aware constraint may be insufficient to prevent interference. We believe characterizing which task pairs exhibit favorable geometry is valuable future work. We have added a discussion of this generalization potential and its conditions to the Conclusion of the revised manuscript.

---

### Review · Reviewer_mYa8 · 2025-12-09

**Summary Of Contributions:**

The paper first present the discory that fine-tuning models preserve the geometric structure of the loss landscapes concerning harmful content, regardless of the fine-tuning method employed, which indicates that fine-tuning shifts but doesn’t erase safety-related geometry in LLMs’ loss landscapes. Building upon this, the author propose to restore safety by using a curvature-aware influence update, which approximates the inverse Hessian on retain data over LoRA parameters and ascend loss on harmful inputs while constraining utility loss. Experiments on different types of LLMs show the proposed method can lower the harmful response rates while preserving the task performance and few-shot ability.

Strength:
1. The paper is well-written and easy to follow.
2. The proposed method are based on some interesting empirical findings, making it more convincing.
3. The evaluation are comprehensive and entails different models and benchmarks.

Weaknesses:
1. Some of the experimental setups are not very clear.
2. The chosen methods for comparison could be improved.
3. The implementation includes several approximations, hyperparameters, and tricks, but their roles within the overall framework are unclear.
Refere to Requested Changes for more details.

**Audience:**

Yes

**Audience Explanation:**

The finding that fine-tuned models preserve the geometric structure of loss landscapes with respect to harmful content is very interesting and could motivate deeper theoretical study. The proposed method enhances LLM safety with minimal performance compromise, demonstrating strong practical value.

**Claims And Evidence:**

Yes

**Claims Explanation:**

The method is motivated by the empirical finding that fine-tuned models preserve the geometric structure of loss landscapes with respect to harmful content. The authors use statistical analysis and loss-landscape visualizations to illustrate this phenomenon. The proposed method is evaluated on comprehensive benchmarks covering safety, accuracy, generalizability, and robustness, and the experiments include several types of LLMs.

**Requested Changes:**

1. In the empirical analysis of section 2, the finetuning process of the model are not clear. Since the authors claim "fine-tuned models preserve the geometric structure of their loss landscapes concerning harmful content, regardless of the finetuning method employed", I would expect to the the results of different finetuning techniques (full-fintuning, lora, SFT, GRPO etc).
2. The current empirical analysis cannot fully convince me. It is unclear whether the preserved loss lanscape is due to the harmful nature of the data, or simply because the content show significant difference to the finetuning data. It can help if the authors could provide the results when the model is finetuned on general-purpose data and evaluate on some domain-specific data, like medical and finance.
3. The connection between the empirical findings and the proposed method are not very clear. Without the findings in Sec. 2, the proposed optimization problem defined in Eq. 2 still make sense. The authors should explain more on why the findings in Sec. 2 serves as an important prerequisite to optimize Eq.2.
4. The proposed algorithm requires a forget set composed of harmful examples. However, most of the comparison baselines do not requires such harmful examples. This makes unfair comparison. The authors should compare with more baselines requiring similar data, e.g., unlearning-based methods.
5.  How sensitive are results to the choice and size of the forget set? How is $\epsilon$ in Eq.2 determined?

---

> ### Author Response · Authors · 2026-01-08
> **Response to Reviewer mYa8**
>
> We thank Reviewer mYa8 for the detailed and constructive feedback, and for recognizing the interesting empirical findings and comprehensive evaluation. We address each concern below.
>
> ### Generalization Across Fine-tuning Methods
>
> > "I would expect to see the results of different finetuning techniques (full-finetuning, lora, SFT, GRPO etc)."
>
> We address this concern in **Common Concern 1** above. Tables R1 and R2 demonstrate that both the geometric preservation finding and our restoration method generalize to full fine-tuning.
>
> ### Out-of-Distribution vs. Harmful Content Preservation
>
> > "It is unclear whether the preserved loss landscape is due to the harmful nature of the data, or simply because the content shows significant difference to the finetuning data."
>
> We directly address this in **Common Concern 1** (Table R1). Domain-specific datasets (CodeAlpaca, MedMCQA, SquAD v2) show highly variable correlations (-0.47 to 0.86) across models, while harmful content shows consistently high correlations (0.85-0.99). If preservation were simply an out-of-distribution effect, all non-training data would show similar patterns. The observed variability confirms safety-specific preservation.
>
> ### Connection Between Section 2 and Method
>
> > "The connection between the empirical findings and the proposed method are not very clear. Without the findings in Sec. 2, the proposed optimization problem defined in Eq. 2 still makes sense."
>
> We thank the reviewer for this opportunity to clarify the crucial connection between our empirical findings and the proposed method.
>
> The loss correlation analysis in Section 2 establishes why small-update restoration is feasible. If safety-relevant loss structure were destroyed during fine-tuning, restoring safety would require large parameter changes that risk catastrophic forgetting on task performance. The high correlation (>0.85) we observe indicates that safety behaviors remain structurally intact but shifted to less influential regions. This means we can navigate back to appropriate refusal behavior rather than reconstruct it from scratch.
>
> This finding directly justifies our constrained optimization in Eq. 2: the constraint L_retain(θ) ≤ L_retain(θ_tuned) + ε with small ε is achievable precisely because safety-relevant regions remain nearby in parameter space. Concretely, the high correlation (>0.85) means we can achieve safety restoration with small ε values (e.g., ε=0.1). Consider the counterfactual: if the correlation were low (say, 0.3), indicating that safety geometry was destroyed, no feasible solution would exist to Eq. 2 with small ε. We would need either (a) large ε causing utility degradation, or (b) to search far in parameter space causing instability. The preserved geometry makes our constrained optimization problem both feasible and stable, and this is the critical connection between Section 2's empirical findings and our method's success. We have added a motivation paragraph at the beginning of Section 3 to better connect the empirical analysis with our method in the revised manuscript.
>
> ### Comparison with Unlearning Methods
>
> > "The authors should compare with more baselines requiring similar data, e.g., unlearning-based methods."
>
> We address this concern in **Common Concern 3** above (Table R4). We also acknowledge that our method requires harmful data while some baselines (SafeLoRA) do not. However, we use only 50 samples compared to 138 for SaLoRA, and for practitioners with access to harmful examples, our method provides substantially better safety outcomes.
>
> ### Hyperparameter Sensitivity
>
> > "How sensitive are results to the choice and size of the forget set? How is ε in Eq.2 determined?"
>
> We address this concern in **Common Concern 2** above (Table R3). Results show robustness across ε ∈ [0.1, 0.4], δ ∈ [0.05, 0.2], and forget set sizes from 50-200 samples.

---

### Review · Reviewer_fsuT · 2025-12-30

**Summary Of Contributions:**

This paper addresses safety degradation in fine-tuned LLMs and proposes a curvature-aware restoration method. Key contributions: (1) an empirical finding that loss landscape geometry is preserved for harmful inputs but shifts for benign data post-fine-tuning; (2) a constrained optimization approach using influence functions and L-BFGS to restore safety while preserving utility; (3) experiments across multiple LLMs showing safety recovery without harming task performance.

**Additional Comments:**

Overall, this is a solid paper with useful contributions. I recommend acceptance provided the above concerns are adequately addressed.

**Audience:**

Yes

**Audience Explanation:**

This paper has several strengths:

1. The geometric preservation observation offers a novel perspective on safety alignment.
2. The method combines established techniques (influence functions, L-BFGS) into a practical, LoRA-compatible pipeline.
3. Comprehensive experiments across multiple models with diverse benchmarks and robustness tests.

I think the community will be interested in this paper.

**Broader Impact Concerns:**

No ethical concerns

**Claims And Evidence:**

Yes

**Claims Explanation:**

This paper provides sufficient arguments and experiments to validate the claims.

**Requested Changes:**

1. Provide direct computational comparisons. While Table 5 reports runtime for the proposed method, a side-by-side comparison of total runtime and memory usage with baselines (especially training-free methods like SafeLoRA) would help practitioners assess trade-offs. A brief cost-benefit analysis for large-scale or continuous fine-tuning scenarios would be valuable.

2. Include sensitivity analyses for theoretical assumptions. The influence function framework assumes small parameter changes and local convexity. Consider adding experiments that vary key hyperparameters (e.g., the utility constraint $\epsilon$ in Eq. 2, or trust region radius $\delta$) to clarify when the second-order approximation remains valid and when it may break down.

3. Results are confined to the LoRA parameter subspace. All experiments are conducted within a LoRA-based fine-tuning framework, making it unclear whether the observed preservation of safety geometry and the effectiveness of the restoration method extend to full fine-tuning or other parameter-efficient adaptation schemes. More experiments are encouraged.

---

> ### Author Response · Authors · 2026-01-08
> **Response to Reviewer fsuT**
>
> ## Response to Reviewer fsuT
>
> We thank Reviewer fsuT for the positive assessment and recognition of our key contributions: the geometric preservation observation, the practical LoRA-compatible pipeline, and comprehensive experiments. We address each concern below.
>
> ### Computational Cost Comparison
>
> > "A side-by-side comparison of total runtime and memory usage with baselines would help practitioners assess trade-offs."
>
> We provide a comprehensive computational comparison measured on the same hardware (H100 GPU) on LLaMA-2-7b-chat-hf:
>
> **Table R5: Computational Comparison**
> | Method | Time (min) | Memory (GB) | Inference Overhead | Harmful Data Required |
> |--------|------------|-------------|--------------------|-----------------------|
> | LoRA (baseline) | 14.3 | 22.0 | None | No |
> | SafeLoRA | 15.0 | 22.0 | None | No |
> | SaLoRA | 18.4 | 34.2 | +10% | Yes (138 samples) |
> | **Ours** | 18.5 | 48.9 | None | Yes (50 samples) |
> | Vaccine | 26.1 | 24.6 | None | No |
>
> Our method requires 18.5 minutes total (3 iterations as specified in Appendix C.1) with 48.9GB peak memory. The higher memory consumption compared to SafeLoRA and SaLoRA is due to the curvature estimation via L-BFGS, which maintains a history of gradient differences. Importantly, this memory overhead occurs only during the one-time restoration phase, not during inference or deployment. Our method introduces no inference-time overhead, unlike SaLoRA, which requires an additional safety module during deployment, and uses fewer harmful samples (50 vs 138). Compared to the Vaccine (26.1 min), our method is faster while achieving substantially better safety restoration, as shown in the main results. This represents a memory-for-accuracy tradeoff: our one-time ~2.2× memory overhead during restoration yields substantially better safety (3.0% vs 11.0% HRR) with no inference overhead. We have added a discussion of these computational trade-offs to Appendix D.6 (Computation Cost) in the revised manuscript.
>
> ### Hyperparameter Sensitivity
>
> > "Consider adding experiments that vary key hyperparameters (e.g., the utility constraint ε in Eq. 2, or trust region radius δ) to clarify when the second-order approximation remains valid."
>
> We address this concern in **Common Concern 2** above, where we provide comprehensive ablations on ε, δ, and forget set size (Table R3). The results demonstrate stable performance across practical hyperparameter ranges.
>
> ### Generalization Across Fine-tuning Methods
>
> > "Results are confined to the LoRA parameter subspace... making it unclear whether the observed preservation of safety geometry and the effectiveness of the restoration method extend to full fine-tuning."
>
> We address this concern in **Common Concern 1** above, where we demonstrate that (1) loss correlations remain high (0.85-0.99) for harmful content under full fine-tuning (Table R1), and (2) our restoration method reduces HRR by 57-75% on fully fine-tuned models while preserving task performance (Table R2).

---

### Author Response · Authors · 2026-01-08
**General response for all Reviewers part 1/3**

We sincerely thank all reviewers for their thoughtful and constructive feedback. We appreciate the recognition of our contributions and have conducted additional experiments to address the raised concerns comprehensively. Below, we first address common concerns shared across reviewers; individual concerns are addressed in each reviewer's thread.

## Common Concern 1: Generalization Across Fine-tuning Methods

**Reviewers fsuT and mYa8** raised important questions about whether our empirical findings (Section 2) and restoration method generalize beyond LoRA to other fine-tuning approaches.

> **fsuT:** "Results are confined to the LoRA parameter subspace... making it unclear whether the observed preservation of safety geometry and the effectiveness of the restoration method extend to full fine-tuning."

> **mYa8:** "I would expect to see the results of different finetuning techniques (full-finetuning, lora, SFT, GRPO etc)." and "It is unclear whether the preserved loss landscape is due to the harmful nature of the data, or simply because the content shows significant difference to the finetuning data."

We appreciate these suggestions and have extended our analysis along two dimensions: (1) loss correlation analysis across fine-tuning methods and data types, and (2) safety restoration effectiveness under full fine-tuning.

### 1.1 Loss Correlation Analysis

We evaluated loss correlations (quantified via Pearson correlation on output responses) for both LoRA and full fine-tuning. To address Reviewer mYa8's question about whether preservation is specific to harmful content or a general out-of-distribution effect, we measured correlations on harmful content (HEx-PHI) as well as diverse domain-specific datasets including code (CodeAlpaca), medical (MedMCQA), and question answering (SquAD v2).

**Table R1: Loss Correlation Across Fine-tuning Methods and Data Types**
| Fine-tuning | Model | Harmful | Dolly | Alpaca | CodeAlpaca | MedMCQA | SquAD v2 |
|-------------|-------|---------|-------|--------|------------|---------|----------|
| **LoRA** | LLaMA-2 7B | **0.992** | 0.056 | -0.055 | 0.551 | -0.465 | 0.596 |
| | LLaMA-3.1 8B | **0.995** | 0.550 | 0.510 | 0.780 | 0.554 | 0.516 |
| | Qwen 2.5 7B | **0.994** | 0.014 | 0.067 | 0.696 | 0.862 | 0.515 |
| **Full FT** | LLaMA-2 7B | **0.852** | -0.004 | 0.185 | 0.396 | 0.597 | 0.649 |
| | LLaMA-3.1 8B | **0.990** | 0.535 | 0.508 | 0.746 | 0.499 | 0.366 |
| | Qwen 2.5 7B | **0.941** | 0.526 | 0.129 | 0.363 | 0.012 | 0.312 |

Harmful content shows consistently high correlation (0.85–0.99) across both LoRA and full fine-tuning, validating that loss structure preservation holds regardless of fine-tuning method. In contrast, domain-specific data shows highly variable correlations ranging from -0.47 to 0.86. While some domain datasets show moderate correlations (e.g., CodeAlpaca 0.55-0.78 for LLaMA-3.1, MedMCQA 0.86 for Qwen), this high variability across datasets and models contrasts sharply with the consistently high correlations for harmful content (0.85-0.99 across all conditions). If preservation were simply an out-of-distribution effect, domain-specific data would show similarly consistent patterns. The observed variability confirms that safety behaviors occupy a functionally distinct region in parameter space.

We note that the loss landscape geometry is visualized in Figure 2 (main paper) for full fine-tuning and Figure 5 (Appendix) for LoRA fine-tuning, providing complementary visual evidence to the quantitative correlations above. We have added these new results to the revised manuscript, Section 2.1.

### 1.2 Safety Restoration Under Full Fine-tuning

We applied our restoration method to fully fine-tuned models to verify that the approach generalizes beyond LoRA:

**Table R2: Safety Restoration Results for Full Fine-tuning**
| Model | Method | HRR ↓ | TruthfulQA | ToxiGen | ARC-C | BoolQ |
|-------|--------|-------|------------|---------|-------|-------|
| LLaMA-2 7B | Original (full FT) | 52.0 | 33.2 | 56.0 | 21.4 | 62.3 |
| | + Ours | **22.5** | 34.5 | 56.8 | 20.7 | 62.1 |
| LLaMA-3.1 8B | Original (full FT) | 42.0 | 36.9 | 42.8 | 51.6 | 83.1 |
| | + Ours | **10.5** | 45.4 | 46.9 | 53.3 | 84.2 |

Our method reduces HRR by 57% (LLaMA-2 7B) and 75% (LLaMA-3.1 8B) while maintaining or slightly improving task performance across all benchmarks. This demonstrates that our curvature-aware approach effectively generalizes to full fine-tuning scenarios. This full fine-tuning result has been added to the Ablation study in Appendix D, section D.1, in the revised manuscript.

---

> ### Author Response · Authors · 2026-01-08
> **General response for all Reviewers part 2/3**
>
> ## Common Concern 2: Hyperparameter Sensitivity
>
> **Reviewers fsuT and mYa8** requested sensitivity analyses for key hyperparameters.
>
> > **fsuT:** "Consider adding experiments that vary key hyperparameters (e.g., the utility constraint ε in Eq. 2, or trust region radius δ) to clarify when the second-order approximation remains valid."
>
> > **mYa8:** "How sensitive are results to the choice and size of the forget set? How is ε in Eq.2 determined?"
>
> We thank the reviewers for highlighting the importance of understanding our method's operating regime. We conducted comprehensive ablations on Llama3.1-8B-Instruct across three hyperparameters: the retain loss threshold ε, trust region radius δ, and forget set size.
>
> **Table R3: Hyperparameter Sensitivity Analysis**
>
> | ε (Retain Loss Threshold) | HRR ↓ | ARC-C | BoolQ |
> |---------------------------|-------|-------|-------|
> | 0.1 | 5.5 | 52.1 | 84.0 |
> | 0.2 | 3.0 | 52.1 | 84.0 |
> | 0.3 | 3.0 | 52.1 | 84.2 |
> | 0.4 | 2.0 | 52.4 | 84.2 |
>
> | δ (Trust Region Radius) | HRR ↓ |
> |-------------------------|-------|
> | 0.01 | 7.5 |
> | 0.05 | 3.0 |
> | 0.10 | 3.0 |
> | 0.20 | 3.5 |
>
> | Forget Set Size | HRR ↓ | HellaSwag | WinoGrande |
> |-----------------|-------|-----------|------------|
> | 50 | 3.0 | 59.1 | 74.3 |
> | 100 | 3.0 | 59.1 | 73.8 |
> | 150 | 4.0 | 59.1 | 73.8 |
> | 200 | 3.5 | 59.1 | 74.4 |
>
> For the retain loss threshold ε, all tested values in [0.1, 0.4] achieve effective safety restoration (HRR ≤ 5.5%) with stable task performance, where larger ε permits more aggressive restoration. For the trust region radius δ, performance remains stable across [0.05, 0.2], with δ = 0.01 yielding overly conservative updates (HRR = 7.5). The adaptive trust region mechanism described in Algorithm 1 automatically adjusts δ when the quadratic approximation becomes inaccurate. For forget set size, even 50 samples suffice for effective restoration (HRR = 3.0%), demonstrating data efficiency, and task performance remains constant across all tested sizes. Small forget set effectiveness is also important for scenarios where harmful examples are scarce or sensitive.
>
> These results indicate that our method is robust across a practical range of hyperparameters. The default settings (ε = 0.1, δ = 0.1, 50 samples) generalize well across all three model families tested in our experiments. We have added a new subsection, Section 4.5 (Hyperparameter Sensitivity), to the Experiments section of the revised manuscript to present the results of this analysis.

---

> > ### Author Response · Authors · 2026-01-08
> > **General response for all Reviewers part 3/3**
> >
> > ## Common Concern 3: Comparison with Unlearning Methods
> >
> > **Reviewers whRS and mYa8** suggested comparing with unlearning-based approaches.
> >
> > > **whRS:** "There is an obvious baseline missing: just optimize L_retain and L_forget together (e.g., min_θ L_retain(θ) − αL_forget(θ))."
> >
> > > **mYa8:** "The authors should compare with more baselines requiring similar data, e.g., unlearning-based methods."
> >
> > We appreciate this suggestion, as it helps clarify the distinction between unlearning and safety restoration. We evaluated standard unlearning methods including Gradient Ascent (θ ← θ + η∇L_forget) and GradDiff with various α values (θ ← θ − η(∇L_retain − α∇L_forget), which corresponds to the baseline min_θ L_retain − αL_forget suggested by Reviewer whRS) on LLaMA-3.1 8B:
> >
> >
> > **Table R4: Comparison with Unlearning Methods**
> > | Method | Forget Loss ↑ | HRR ↓ | TruthfulQA | ToxiGen | ARC-C | BoolQ |
> > |--------|---------------|-------|------------|---------|-------|-------|
> > | Original (fine-tuned) | 8.8 | 25.5 | 43.6 | 46.0 | 51.8 | 84.0 |
> > | Gradient Ascent | 25.9 | 24.5 | 34.8 | 43.6 | 30.5 | 73.8 |
> > | GradDiff (α=0.5) | 44.0 | 22.0 | 34.5 | 55.7 | 20.2 | 43.6 |
> > | GradDiff (α=1.0) | 19.0 | 27.0 | 36.5 | 43.3 | 18.8 | 55.8 |
> > | GradDiff (α=2.0) | 17.9 | 23.8 | 35.5 | 43.0 | 47.5 | 82.1 |
> > | **Ours** | **15.5** | **3.0** | 45.4 | 46.9 | 53.3 | 84.2 |
> >
> >
> > **The forget loss column reveals a critical insight: high loss does not guarantee safety.** GradDiff with α=0.5 achieves the highest forget loss (44.0), 5× the original model (8.8), yet provides minimal HRR improvement (22.0% vs 25.5%). Most strikingly, our method achieves comparable forget loss to GradDiff (α=2.0) (15.5 vs 17.9) but dramatically different safety outcomes (3.0% vs 23.8% HRR), an 8× improvement despite similar loss values. This observation aligns with recent findings [1], which demonstrate that untargeted unlearning methods produce unpredictable outputs—often gibberish or hallucinations rather than coherent responses. They note that "higher loss can result from random token sequences or incoherent text rather than the coherent refusals desired for safety," which directly explains why high forget loss in Table R4 does not translate to improved safety outcomes. These results reflect fundamental differences in objectives and mechanisms.
> >
> > First, unlearning and restoration address different goals. Unlearning aims to forget specific behaviors by removing the model's capability on certain data, whereas our goal is to restore the refusal behavior that was suppressed during fine-tuning. These are conceptually distinct: unlearning removes knowledge, while restoration recovers latent alignment that remains structurally preserved (as shown in Table R1).
> >
> > Critically, unlearning methods optimize for higher loss on harmful data, while our goal is to restore appropriate refusal responses like "I cannot assist with that request." Simply maximizing loss often produces incoherent outputs (random tokens, gibberish) rather than helpful, safe refusals, where the two objectives are fundamentally different.
> >
> > Third, effective unlearning typically requires large parameter updates to significantly alter the output distribution. In our experiments, we constrained all methods to operate within similar update budgets (controlled via ε) for fair comparison. Under these constraints, first-order unlearning methods lack the precision to selectively restore safety without degrading utility. As discussed in Appendix D.2 (Figures 8-9), first-order methods exhibit oscillatory behavior in the optimization landscape, which explains the degraded task performance observed in Table R4.
> >
> > These results suggest that safety restoration requires a more nuanced approach than standard unlearning, supporting the value of curvature-aware optimization that leverages the preserved loss structure to navigate toward appropriate refusal behavior while respecting task-relevant regions. To address this concern, we have revised Appendix D.3 (Connection to Machine Unlearning) to include a new result and additional clarification.
> >
> >
> > **Reference:**
> >
> > [1] A Closer Look at Machine Unlearning for Large Language Models (ICLR 2025)

---

### Author Response · Authors · 2026-01-08
**Summary of Manuscript Revisions**

Based on reviewer feedback, we have made the following revisions:

1. **Section 2.1**: Added Table R1 (loss correlations across fine-tuning methods and data types)
2. **Section 3**: Added motivation paragraph connecting empirical findings to optimization approach
3. **Section 4.5** (new): Hyperparameter sensitivity analysis (Table R3)
4. **Appendix D.1** (new): Full fine-tuning restoration results (Table R2)
5. **Appendix D.3**: Expanded comparison with unlearning methods (Table R4)
6. **Appendix D.5** (new): Quantitative analysis of loss landscape preservation (Table R6)
7. **Appendix D.6**: Computational cost comparison (Table R5)
8. **Conclusion**: Discussion of generalization to arbitrary task pairs

We thank all reviewers again for their constructive feedback, which has substantially strengthened our paper.

---

### Decision · Action_Editor_Bjqs · 2026-02-10

**Recommendation:** Accept as is

**Audience:**

Yes

**Audience Explanation:**

Many researchers work in this field.

**Claims And Evidence:**

Yes

**Claims Explanation:**

This paper found that safety-related loss-landscape geometry remains after downstream fine-tuning and uses this insight to propose a new method which preserves curvature with influence/second-order updates to selectively raise loss on harmful inputs. The rebuttal provided extensive updates especially to the experiments. All reviewers were satisfied with the update and recommended acceptance.